

# Modeling of aerosol property evolution during winter haze episodes over a megacity cluster in northern China: Roles of regional transport and heterogeneous reactions

Huiyun Du[1,2], Jie Li[1,2,3*], Xueshun Chen[1], Zifa Wang[1,2,3], Yele Sun[1,2,3], Pingqing Fu[1],
Jianjun Li[4], Jian Gao[5], Ying Wei[1,2]

[1] LAPC, Institute of Atmospheric Physics, Chinese Academy of Sciences, Beijing 100029, China

[2] College of Earth Sciences, University of Chinese Academy of Sciences, Beijing 100029, China

[3] Center for Excellence in Urban Atmospheric Environment, Institute of Urban Environment, Chinese Academy of Sciences, Xiamen, China

[4] China National Environmental Monitoring Center, Beijing, China

[5] Chinese Research Academy of Environmental Sciences

**Abstract.** Regional transport and heterogeneous reactions played crucial roles in haze formation over a megacity cluster centered on Beijing. In this study, the updated Nested Air Quality Prediction Model System (NAQPMS) and the HYSPLIT Lagrangian trajectory model were employed to investigate the evolution of aerosols—in terms of the number concentration, size distribution, and aging degree—in Beijing during six haze episodes between November 15 and December 15, 2016, as part of the Air Pollution and Human Health–Beijing (APHH-Beijing) winter campaign of 2016. The model exhibited reasonable performance not only for mass concentrations of $PM_{2.5}$ and its components in Beijing but also for the number concentration, size distribution, and aging degree. We discovered that regional transport played a nonnegligible role in haze episodes, with contributions of 14%–31% to the surface $PM_{2.5}$ mass concentration. The contribution of regional transport to secondary inorganic aerosols was larger than that of regional transport to primary aerosols (30%–63% vs. 3%–12%). The chemical transformation of $SO_2$ in the transport pathway from source regions to Beijing was the major form of $SO_4^{2-}$ regional transport. We also found that sulfate formed outside Beijing from $SO_2$ that was emitted in Beijing; this sulfate was then blown back to Beijing and considerably influenced haze formation. In the transport pathway, aerosols



underwent aging, which altered the mass ratio of coating to black carbon ($R_{BC}$) and the
size distribution of number concentrations. During the episodes, the geometric mean
diameter (GMD) increased from less than 100 nm at the initial site to approximately
120 nm at the final site (Beijing), and $R_{BC}$ increased from 2–4 to 4–8. These changes
would affect regional radiation and climate. In haze episodes with high humidity, the
average contributions of gas and aqueous chemistry, heterogeneous chemistry, and
primary sulfate emission were comparable. Primary emissions had the greatest impact
under light to moderate pollution levels, whereas heterogeneous chemistry had a
stronger effect under high pollution levels.
**Keywords**: Regional transport; heterogeneous reactions; number size distribution;
NAQPMS

# 1 Introduction

In past decades, a megacity cluster in China that is centered on Beijing and
includes 28 cities (272,500 $km^2$, a population of 191.7 million people) has been
experiencing frequent severe and persistent haze episodes (Zhao et al., 2013; Sun et al.,
2014; Sun et al., 2016). $PM_{2.5}$ levels exceeding 500 μg $m^{-3}$ have often been reported.
The adverse effects of $PM_{2.5}$ on visibility, climate, and particularly human health have
drawn widespread public attention (Hyslop, 2009; Chen et al., 2018; Yang et al., 2017a;
Yang et al., 2017b; Anderson et al., 2010). Although the $PM_{2.5}$ concentration in Beijing
has decreased by 35% in the recent 5 years (2013–2017) because of implementation of
the Atmospheric Pollution Prevention and Control Action Plan, the $PM_{2.5}$ level in
Beijing in 2017 still reached 58 μg $m^{-3}$, which is 1.7 times the World Health
Organization-recommended      safe      level      of      35      μg      $m^{-3}$
(http://www.bjepb.gov.cn/bjhrb/index/index.html). Understanding the mechanism of
haze episodes in this megacity cluster is thus an urgent task for policymakers.
Observations have revealed that haze episodes in this megacity cluster are mainly
caused by the rapid formation of secondary inorganic species (SIA, including sulfate,
nitrate, and ammonium) (Huang et al., 2014; Zheng et al., 2015; Han et al., 2016). The



SIA mass fraction in PM$_{2.5}$ can be up to 55% on days of severe pollution, which is 2.5
times that on clear days (Ma et al., 2017). Tang et al. (2016) proposed that local
chemical transformation related to humidity dominates the rapid formation of SIA in
Beijing. Yang et al. (2015) argued that local chemical conversion would not be fully
able to support the observed rapid formation of SIA in a short time. Using a ceilometer
and in situ observation data, Zhu et al. (2016) and Ma et al. (2017) further proposed
that regional transport was the major cause of the initial haze stage and that local
chemistry, particularly heterogeneous chemistry, dominated the later rise in Beijing.
This result is different from the findings obtained using numerical air quality models
(LOTOS-EUROS, Regional Air Quality Model System [RAQMS], and the Nested Air
Quality Prediction Modeling System [NAQPMS]) (Timmermans et al., 2017; Li and
Han, 2016; Li et al.,2017), in which regional transport dominated during haze episodes
in the megacity cluster. Recent observations of the physiochemical properties (e.g.,
mixing state, number concentration, and size distribution) of aerosols can provide more
information for improving the accuracy of regional transport and chemistry assessment.
Black carbon (BC) is usually more thickly coated by SIA and organic aerosols in
transported and aged air masses than in fresh particles, as indicated by the higher mass
ratio of coating to BC (R$_{BC}$) (Wang et al., 2018). Massoli et al. (2015) and Wang et al.
(2017) reported that R$_{BC}$ exceeded 10 in remote sites after BC had undergone long-term
transport. This value was much higher than that in an urban area with high fresh particle
emissions, where R$_{BC}$ generally was less than 1.5 (Liu et al., 2017). The geometric mean
diameter (GMD) of PM$_{2.5}$ also changed significantly due to the impact of regional
transport. In haze episodes in Beijing, the GMD increased to 120 nm in regionally
transported air masses, which is twice that under clean conditions (Ma et al., 2017).
Investigating the evolution of aerosol properties other than mass concentration during
regional transport is thus essential for assessing the roles of regional and local chemistry.
Such investigations are rarely conducted using the current three-dimensional chemical
transport models, which mostly concentrate on mass concentrations. The treatment of
heterogeneous chemistry is likely another source of modeling uncertainty. The current



models generally account for only a part of the observed $SO_4^{2-}$ concentrations (Wang
et al., 2014). Heterogeneous chemistry is considered critical to improving model
performance (Zheng et al., 2015; Cheng et al., 2016; Li et al., 2018).
From November 15 to December 15, 2016, a field campaign was carried out in
Beijing within the framework of the UK-China Air Pollution & Human Health (APHH)
project. Nearly 30 Chinese and British institutions—including the Institute of
Atmospheric Physics, Chinese Academy of Sciences, Leeds University, the University
of Birmingham, the University of Reading, Tsinghua University, and Peking
University—participated in this campaign. Aerosol properties such as the size
distribution, number concentration, and mixing states were simultaneously measured in
China. APHH-Beijing aimed to explore the sources of and processes affecting urban
atmospheric pollution in Beijing. In the present study, we used the NAQPMS to
simulate aerosol properties in the campaign period as a part of the APHH winter
campaign. To improve model performance, the NAQPMS was updated by coupling it
with an advanced particle microphysics (APM) module that explicitly accounts for the
microphysical process and a new heterogeneous chemistry scheme (Chen et al., 2014;
Li et al., 2018). The hybrid single-particle Lagrangian integrated trajectory model
(HYSPLIT) was also employed to assess the evolution of aerosol properties (e.g.,
mixing state, number concentration, and size distribution). Crucially, the effects of
regional transport and heterogeneous chemistry on aerosol properties were quantified.
To the best of our knowledge, this is the first study to distinguish the contributions of
transport of SIA itself and its precursors to $PM_{2.5}$ in Beijing. We believe that this study
is helpful to understanding the causes of haze in this megacity cluster.

## 2 Model description and methodology

### 2.1 Model description

The Nested Air Quality Prediction Model System (NAQPMS) developed by the
Institute of Atmospheric Physics, Chinese Academy of Sciences (IAP/CAS) is a three-



dimensional Eulerian terrain-following chemical transport model. WRFv3.6.1, driven
by Final Analysis (FNL)data from the National Centers for Environmental Prediction
(NCEP), provides the meteorology field for the NAQPMS. The NAQPMS includes
emission, horizontal and vertical advection and diffusion, dry and wet deposition, and
chemical (including gas, aqueous, and heterogeneous) reaction processes (Wang et al.,
2001; Li et al., 2012; Li et al., 2018). It also incorporates online source tagging, process
analysis, an online WRF coupler, and other techniques (Wu et al., 2017; Wang et al.,
2014). The Carbon Bond Mechanism version-Z (CBMZ) is used for gas-phase
chemistry mechanisms. The thermodynamic model ISORROPIAI1.7 is used to
calculate the composition and phase state of an $NH_4^+$–$SO_4^{2-}$–$NO_3^-$–$Cl^-$–$Na^+$–$H_2O$
inorganic aerosol system (Nenes et al., 1998). Six secondary organic aerosols are
managed using a two-product module. Further details of the NAQPMS can be found in
the studies of Li et al. (2013, 2014, 2017), and numerous subsequent papers have been
published describing recent updates.

To accurately describe aerosol properties (e.g., number concentration, size

distribution, and mixing states), an advanced multitype, multicomponent, size-resolved
microphysics (APM) module is coupled to the NAQPMS (Chen et al., 2014). APM
explicitly describes microphysical aerosol processes, including nucleation,
condensation, evaporation, coagulation, thermodynamic equilibrium with local
humidity, hygroscopic growth, and dry and wet deposition (Yu and Luo, 2009), and it
has already been applied in the global GEOS-Chem model (Ma et al., 2014). In the
updated NAQPMS, 40 sectional bins covering 0.0012–12 μm were used to represent
secondary particle distribution ($SO_4^{2-}$, $NO_3^-$, $NH_4^+$, and secondary organic aerosols)
(Chen et al., 2014). The size distribution of BC and primary organic aerosol was
represented using 28 section bins. Other primary particles such as dust and sea salt were
represented using four bins. The coating of secondary species on primary particles (sea
salt, BC, OC, and dust) was explicitly simulated using a scheme that dynamically
calculates the aerosol aging time with an hourly resolution on the basis of aerosol
microphysics. The mass concentrations of coating species were also tracked in the





model. Chen et al. (2017) employed the updated NAQPMS and revealed that the
daytime aging time of BC in Beijing in winter can be less than 2 hours. This is much
less time than the fixed aging time scale of 1.2 days that has been stipulated in previous
studies (Liu et al. 2009) but is close to observed levels (2–4 hours) (Peng et al. 2016).
Li et al. (2018) further developed a heterogeneous chemical scheme based on mixing
states to reproduce the chemical transformation of gaseous precursors on an aerosol
surface, which largely altered the sizes and hygroscopicity of particles. Comparison
with long-term observations has proven that the updated NAQPMS can successfully
estimate aerosol mass and the number concentration, size distribution, mixing states,
and BC aging time in China (Li et al., 2017, 2018; Chen et al., 2014, 2017).

Distinguishing the contributions of the transport of SIA itself and its precursors to

$PM_{2.5}$ is always difficult (Sun et al., 2014; Li et al., 2014, 2017; Ying et al., 2014). These
contributions have generally been named regional transport in studies; this leads to
ambiguity in regional transport. In this study, we divided the secondary species (e.g.,
SIA) in the $i^{th}$ receptor region into four parts: 1) SIA locally produced from the $i^{th}$ locally
emitted precursors (LC); 2) SIA chemically formed in other regions from the $i^{th}$ locally
emitted precursors (LTC); 3) SIA chemically formed in the transport pathway to the $i^{th}$
receptor region of precursors emitted in the $j^{th}$ source region (RTC); and 4) SIA
produced in the $j^{th}$ region from precursors emitted in the $j^{th}$ source region (RLC).

An online tracer-tagging module in the NAQPMS was used to resolve the

contributions from LC, LTC, RTC, and RLC. The module is capable of tracing both the
emission regions of precursors and the formation regions of secondary aerosols. First,
the mass contribution from the locations in which SIA was formed, called $C_2$, was
tagged. The mass contribution from precursors emitted in different locations, called $C_1$,
was then tagged. More technical details can be found in the studies of Li et al. (2014)
and Wu et al. (2017). The following equation can be employed to calculate the degree
of chemical conversion during transport (TC):

$$\text{TC} = \sum_{i=1}^{n}(C_{1i} - C_{2i} \times CC_i) \qquad (1)$$

where $C_{1i}$ refers to the absolute mass concentration transported to the receptor site,





produced by precursors emitted in region $i$;
$C_{2i}$ refers to the absolute mass concentration formed in region $i$ and transported to
receptor site;
$CC_i$ refers to the local contribution ratio of precursors in region $i$;
$C_{2i} \times CC_i$ refers to the absolute mass transported to receptor site and generated at
region $i$ by chemical conversion of precursors released at region $i$. When $i = 1$, it refers
to LC; when $i \neq 1$, $\sum_{i=2}^{n} C_{2i} \times CC_i$ refers to RLC;
$C_{1i} - C_{2i} \times CC_i$ is the mass concentration generated in all regions except $i$ by chemical
conversion of the precursors released at region $i$ and finally transported to the receptor
site. When $i = 1$, it refers to LTC; when $i \neq 1$, $\sum_{i=2}^{n}(C_{1i} - C_{2i} \times CC_i)$ refers to RTC.

In this study, 10 regions according to administrative division are selected for

source tagging (Fig. 1c), six of which—Chengde, Zhangjiakou, and Qinhuangdao
(NHB); Beijing (BJ); Tianjin (LT); Hengshui, Xingtai, and Handan (SHB); Baoding
and Shijiazhuang (WHB); and Tangshan, Langfang, and Cangzhou (EHB)—are a part
of the Beijing–Tianjin–Hebei (BTH) area. Henan (HN), Shandong (SD), Shanxi (SX),
and other regions (OT) are regions outside the BTH area.
**2.2 Model structure**

Simulation was conducted from November 10 to December 15, 2016, and the first

5 days were set aside as a spin-up period. The three nested model domains were shown
in Fig. 1a. The horizontal resolutions were 27, 9, and 3 km from the coarsest to
innermost domain. The first level of the NAQPMS was approximately 20 m in height,
and there were approximately 17 layers under 2 km.

To quantitatively assess the contribution of primary emissions, traditional

chemistry reactions (gas-phase and aqueous chemistry reactions), and heterogeneous
chemistry to sulfate, three sensitivity simulations were conducted. The baseline
scenario (Base) involving all heterogeneous reactions considered primary sulfate
emissions and its results were used for model verification and source apportionment
analysis. Control 1 (C1) involved all heterogeneous reactions but did not consider





primary sulfate emissions. Compared with Base, Control 2 (C2) excluded the
heterogeneous reactions of $SO_2$. Base–C2 represents the effect of heterogeneous
reactions on sulfate. Base–C1 represents primary sulfate emissions.
The HYSPLIT model was used to analyze the trajectories of air masses (Draxier
and Hess, 1998). The calculated trajectories are helpful to resolving the evolution of
aerosol properties in the transport pathway by extracting the simulated results by the
NAQPMS along trajectories. In this study, the same meteorology data (obtained hourly
data of the third domain) used in the NAQPMS were employed to perform trajectory
analysis; this avoided the errors caused by inconsistency between the two models (the
NAQPMS and HYSPLIT).
**2.3 Emission inventory**
The anthropogenic emissions were obtained from the 0.25° × 0.25° Multi–
resolution Emission Inventory for China (MEIC), and the base year was 2016 for BTH
(http://www.meicmodel.org/publications.html). In addition, observation data collected
at sites within BTH were used to update the MEIC on the basis of their latitude and
longitude information. Biomass burning emissions were taken from the Fire Inventory
from NCAR (National Center for Atmospheric Research) (Wiedinmyer et al., 2011).
Primary sulfate was assumed to constitute 5% of $SO_2$ emissions in the original MEIC
inventory, but through in situ measurement of source profiles, Cao et al. (2014), Wang
et al. (2009), Zheng et al. (2013), and Ma et al. (2015) discovered that primary sulfate
comprised approximately 40%, 6%, and 15% of primary $PM_{2.5}$ from industrial, power,
and residential emissions, respectively, in the main form of $(NH4)_2SO_4$. Thus, we
modified primary sulfate emissions in this study. Figure 1b displays the hourly primary
$PM_{2.5}$ emission rate.
**2.4 Observations**
The surface meteorological parameters were obtained from the China
Meteorological Administration, whereas the vertical profiles of meteorological





parameters   were   obtained   from   the   University   of   Wyoming
(http://weather.uwyo.edu/upperair/sounding.html). Observations of $PM_{2.5}$, $SO_2$, $NO_2$,
and $O_3$ concentrations were obtained from the China National Environmental
Monitoring Center (http://www.cnemc.cn/). Aerosol components (including BC,
organic matters [OM], sulfate, nitrate, and ammonium) were measured in situ by using
an Aerodyne high-resolution time-of-flight aerosol mass spectrometer. Details of the
instruments can be found in the study by Sun et al. (2015). The particle number size
distributions at ground level was obtained using a scanning mobility particle sizer
(SMPS) with a time resolution of 5 min. Details of the instruments can be found in the
study by Du et al. (2017). All data in this study are presented in Beijing local time (UTC
+ 8 h).

## 3 Model validation

### 3.1 $PM_{2.5}$ mass and number concentrations and aging degrees

The time series of simulated and observed $PM_{2.5}$ in different cities of BTH from

November 15 to December 15, 2016, are illustrated in Fig. 2. In the study period, six
regional haze episodes were identified, namely, November 15–20 (Ep1), November 23–
26 (Ep2), November 28–30 (Ep3), December 2–4 (Ep4), December 6–8 (Ep5),
December 10–12 (Ep6). The $PM_{2.5}$ mass concentration frequently exceeded 200 μg m$^{-3}$
and the average concentration reached 120 μg m$^{-3}$ during episodes. Haze usually
formed in several hours; for example, the increasing rate of $PM_{2.5}$ reached 200 μg m$^{-3}$
h$^{-1}$ and lasted approximately 12 hours in Tangshan. These observed haze patterns were
generally reproduced by the NAQPMS. The correlation coefficient (R) between the
observation and simulation in most cities was 0.6–0.8, and 60%−80% of simulation
results were within a factor of 2 of the observation. The mean fractional bias (MFB)
and mean fractional error (MFE) ranged from −0.07 to 0.7, meeting the criteria of MFB
≤ 0.6 and MFE ≤ 0.75 (Boylan et al., 2006). The simulation did however underestimate
$PM_{2.5}$ in Beijing and Baoding for Ep2. This was caused by the failure of the mineral





aerosol transport simulation. Compared with other cities in the cluster, Beijing and
Baoding are closer to the Gobi Desert, a major dust source in East Asia, and they are
thus more easily affected by dust storm transport. Pan et al. (2018) found a pronounced
peak in the size distribution at 4–5 μm for Ep2 in Beijing. The concentrations of $Ca^{2+}$
was 7 times the campaign averages (Fig. S1).
The aerosol components in Beijing, Langfang, and Baoding are compared in Fig.
3. In general, the simulation largely reproduced the variation in primary and secondary
aerosols. In particular, the rapid increase in SIA during Ep1, Ep2, and Ep4 was captured
by the simulation. Interestingly, the NAQPMS underestimated the sulfate concentration
in Beijing during Ep2 and Ep4, but the nitrate and ammoniate concentrations during
these two episodes were successfully reproduced. This was related to the transport of
mineral dust (Ep2) and local emissions (Ep4). As discussed in the last paragraph,
Beijing had high mineral loadings for Ep2, which provided a favorable medium for
chemical transformation of anthropogenic $SO_2$ into sulfate in the form of $CaSO_4$ or
$MgSO_4$ (Zhuang et al., 1992). Underestimation of the sulfate concentration for Ep4 may
have been caused by local emissions in Beijing. As illustrated in Fig. 3, the simulation
failed to reproduce the sharp increase in both sulfate and BC in Beijing during this
episode. This is different from the case of Ep2, in which sulfate was underestimated but
BC was favorably reproduced. Wang et al. (2009) and Ma et al. (2015) found that sulfate
accounted for 40% and 6.6% of primary $PM_{2.5}$ emissions from industry and power
plants, which also emit a large amount of BC. This sharp increase in BC was a local-
scale episode. In Langfang, a site 50–60 km from Beijing, both the observed and
simulated BC concentration increased slowly to 20 μg m$^{-3}$, which is much less than that
in Beijing (45 μg m$^{-3}$). The monthly emissions employed in this study made it difficult
to capture these short-term local-scale emission changes. The simulated $SO_2$ and $NO_2$
concentrations are compared with the observations in Fig. S2, and the normalized mean
bias (NMBs) of these concentrations were less than 40%.
The number size distribution is critical to examining aerosol evolution during haze
episodes (Ma et al., 2017). In this study, both the simulation and observation revealed



a rapid increase in the GMD from 50 to approximately 120 nm during the initial stages
of episodes in Beijing (Fig. 4). The observed mean number concentration of aerosols
(dN/dlogDp) showed a unimodal distribution and was mainly concentrated in the
Aitken mode (25–100 nm) and accumulation mode (100–1,000 nm). The highest
concentration was approximately $1.8 \times 10^4 \ cm^{-3}$ at a 100-nm diameter. These patterns
were favorably reproduced by the simulation. The simulated number concentrations
were underestimated in 10–60 nm by 20%–30% and overestimated in 80–150 nm by
20%. This indicated that the model needs to be improved regarding its treatment of new
particle formation and the volatility of primary organic aerosols.
Herein, the aging degree of BC is represented by the mass ratio of coating to BC
($R_{BC}$), which has been widely used in previous studies (Oshima et al., 2009; Collier et
al., 2018). Figure 11 shows that the mean simulated $R_{BC}$ in Beijing was 4.5 and 5.0 in
the entire study period and during pollution episodes, receptively, which are extremely
close to the observations (~5.0 and 5.1) (Wang et al., 2018). The high performance of
the model in terms of mass and number concentrations, compositions, and the mixing
state of aerosols gives us confidence for analyzing aerosol evolution during transport
in the megacity cluster centered on Beijing.
**3.2 Meteorology**
The simulated wind direction and speed coincided with the observations for the
haze episodes. In particular, the model captured low wind speeds, and the times at which
the wind shifted direction were well reproduced (Fig. S3). Regarding relative humidity
and temperature, WRF performed high values of R (0.68–0.93) and low NMBs (−0.51
to 0.44) (Table S1). In particular, the high relative humidity during Ep1 was well
reproduced. Inversion layers were present during the initial stage of haze formation (Fig.
S4). The height of the inversion layers varied among episodes. During Ep1 and Ep6,
strongly elevated inversion layers were present between 1 and 2 km, whereas the
inversion layers were close to the surface during other episodes. Temperature inversion
is favorable for pollution accumulation, and the model reproduced this feature favorably.



In sum, the high performance of the meteorological simulation gave us confidence for
PM$_{2.5}$ simulation.

## 4 Results and discussion

**4.1 Source apportionment of surface PM$_{2.5}$**

The simulated spatial distribution of average surface PM$_{2.5}$ levels and the wind
vector during the six haze episodes are shown in Fig. 5. In general, two types of patterns
were observed. The first pattern corresponded to Ep1, Ep4, and Ep6 and reflected that
a highly polluted belt with >200 μg m$^{-3}$ PM$_{2.5}$ extended from the southwest to the
northeast along the Taihang mountain range. In the second pattern (Ep2, Ep3, and Ep5),
the PM$_{2.5}$ level of 150–200 μg m$^{-3}$ was concentrated in three northern cities (Beijing,
Tianjin, and Tangshan). In the other cities, the PM$_{2.5}$ mass concentrations ranged from
75 to 115 μg m$^{-3}$, indicating a light pollution level according to the Technical
Regulation on Ambient Air Quality Index (on trial).
Figure 6 shows the contributions of regional transport and local emissions to
average PM$_{2.5}$, primary aerosol (PA, BC and primary PM$_{2.5}$), and SIA levels in different
cities during the study period. The contribution of local emissions was more than that
of regional transport to the PM$_{2.5}$ mass concentration in all cities, except Heng Shui,
Cangzhou, Langfang, and Qinhuangdao; the magnitude of local emission contributions
was 49%–80%. The principle reason for this was the accumulation of local PA
emissions. In most cities, 64%–93% of PA originated from local emissions (Fig. 6c). In
contrast to PA, the SIA contribution was dominated by regional transport of emissions
in other cities (50%–87%). Even the emissions of cities outside the city cluster (e.g.,
Henan, Shanxi and Shandong) were transported to the megacity cluster, travelling 500–
1,000 km. In Beijing, the local contribution to total PM$_{2.5}$ and PA was 74% and 94%,
respectively, whereas regional transport from other cities was the major source of SIA,
contributing 51%. The difference in source apportionment between PA and SIA was
related to the mechanisms of PA and SIA formation. Regarding PA, the inversion layer



and weak winds during stable weather conditions prevented PA transport and resulted
in local-scale accumulation of PA emissions. SIA mostly originated from the chemical
conversion of its gaseous precursors (e.g., $SO_2$, $NO_2$, and $NH_3$). The regional transport
provided sufficient time (1–3 days) and aerosol surface for this chemical transformation
(Li et al., 2015; Li et al., 2017). This also indicates that regional controls would be the
most efficient way to decreasing the SIA concentration in this megacity cluster. Our
results agree favorably with the observed impact of regional emission controls in Asia-
Pacific Economic Cooperation China 2014. During this gathering, the SIA
concentration in Beijing decreased to a greater degree than the PA concentration
because of regional controls (Sun et al., 2016).
The source apportionment in haze episodes in Beijing is illustrated in Fig. 7.
Regional transport contributed 14%–31% to the surface $PM_{2.5}$ mass concentration
during the six episodes. The highest contribution of regional transport occurred in Ep1
and Ep5 (29% and 31% of the total $PM_{2.5}$, respectively). In Ep1 and Ep5, the
contribution of the SIA originating from regional transport reached 53% and 63%,
respectively. Interestingly, the regionally transported SIA had different source regions
in Ep1 and Ep5. In Ep5, SX, WHB, and NHB were the dominant source regions,
whereas the source regions for Ep1 were more diverse. This indicates the complexity
of regional transport in this megacity cluster. Compared with the episodes in November
2015, the effects of regional transport of $PM_{2.5}$ and SIA mass concentrations were
weaker in this study, which may be related to the weather system and emission controls
in 2016 (Li et al., 2017). Therefore, more studies on regional transport should be
conducted to further understand regional haze formation mechanisms. In other episodes
(Ep2, Ep3, Ep4, and Ep6), regional transport of surface $PM_{2.5}$, PA, OM, and SIA mass
concentrations were in the range 14%–23%, 3%–12%, 3%–14%, and 30%–51%,
respectively. Local emissions during the episodes were more dominant than the
monthly averages.
Figure 8 presents the relative contribution of regionally transported SIA under
different levels of pollution in Beijing. The source regions varied considerably under



different pollution levels. Under clean conditions (when SIA < 50 µg m$^{-3}$), NHB and
SX were the main source regions, contributing up to 30% and 19%, respectively. With
the increase of SIA concentrations, WHB, SD, and EHB became the main source
regions, contributing 27%, 15%, and 13%, respectively, which is consistent with
transport along the southwest and southeast corridors of BTH. Under heavy pollution,
pollutants from HN and farther regions were blown to Beijing, resulting in a remarkably
higher contribution of HN. This indicates that wider regional emission control is
necessary to reduce severe pollution.

## 376   4.2 Impact of regional transport of sulfate and its precursors

## 377   on Beijing

Quantifying the impact of regional transport of sulfate and its precursors is a crucial
task. Sun et al. (2014) considered sulfate formed outside Beijing as regionally
transported sulfate, and they estimated that its contribution reached 67% during winter
haze episodes. By tagging emissions regions of precursors in models and ignoring
where secondary aerosols were formed, Li et al. (2017) and Timmermans et al. (2017)
estimated the contribution of transport to be 40%–50%. These estimated contributions
of regional transport are different in physical meaning, which may confuse
policymakers. In this study, we divided the sulfate mass concentration in Beijing into
four parts, LC, LTC, RLC, and RTC as described in Sect. 2.1. The regional transport
defined by Sun et al. (2014) was LTC + RLC + RTC, whereas in the studies by Li et al.
(2017) and Timmermans et al. (2017), it was RLC + RTC. In this study, we employed
RLC + RTC as representing regional transport.
Figure 9a shows the contributions of LC, LTC, RLC, and RTC to the daily average
sulfate concentration in Beijing during the study period. RTC and LC were the
dominant sources of sulfate, contributing 71%–89% in total. The contributions of RTC
ranged from 29% in Ep6 to 59% in Ep2, and contributions of LC were 30%–42%. RTC
dominated the regional transport over the whole period, which indicates that chemical
conversions in the transport pathway of SO$_2$ were critical to haze formation. Notably,





the LTC contribution was comparable with that of LC in Ep3, Ep4, and Ep6. This
suggests that the $SO_2$ emitted in Beijing was blown away and formed sulfate outside of
Beijing. These formed sulfates may have been blown back to Beijing under certain
weather conditions and were previously considered regional transport. The contribution
of LTC also largely explains the difference in estimated regional transport contributions
between Sun et al. (2014) and Li et al. (2017). In the present study, LTC + RLC + RTC
accounted for 58%–70% of the sulfate concentration in the six episodes, which is
relatively similar to the estimation (75%) of Sun et al. (2014), which was based on the
observed hourly rate of increase of local sulfate concentration.

In the initial and subsequent pollution stages, LC, LTC, and RTC showed different

patterns in Beijing. In Ep1, local contributions dominated before the sulfate
concentration increased rapidly (November 15 and 16). In particular, sulfate blown
back to Beijing from its local emissions (LTC) made a larger contribution (35%) than
RTC (25%). In the rapid rising phase of sulfate (November 17 and 18), contribution of
RTC increased from 25% to 47%. LC was also significant and increased considerably
from 37% to 41%. These two parts (LC and RTC) explained the rapid formation of
sulfate in Beijing. This suggests that the joint control of local and regional $SO_2$
emissions is essential for preventing the rapid formation of haze in this region, which
is receiving considerable attention and eliciting widespread interest among the
researchers and policymakers (Sun et al., 2014; Ma et al., 2017; Li et al., 2017). This
feature is also reflected in Fig. 9b. Under clear conditions (sulfate < 20 μg m$^{-3}$), the
local contributions (LC and LTC) were positively correlated with the sulfate mass
concentration. In total, they contributed 40%–60% of the sulfate mass concentration.
The ratio of LC to LTC was approximately 2:1. Under moderate sulfate levels (20 μg
m$^{-3}$ < sulfate < 35 μg m$^{-3}$), the local contribution was lower—particularly the LTC—
leading to a ratio of LC to LTC of approximately 8. Sulfate formed in the regional
transport pathway (RTC) significantly increased from 40 to 65%. Under heavy
pollution levels (> 35 μg m$^{-3}$), the LC was up to at 50% due to extremely stable
boundary layers. Our results are consistent with those of Ma et al. (2017), in which





regional transport and local heterogeneous chemistry were qualitatively discovered to
make high contributions to initial and subsequent pollution stages.

## 4.3 Evolution of aerosol properties in Beijing during haze episodes

Aerosol properties such as the particle size and aging degree can change
dramatically on haze days because of fresh emissions, subsequent chemical conversions,
and regional transport, which strongly affect regional radiation and climate (Cappa et
al., 2012). As illustrated in Fig. 4b, the GMD of aerosols in Beijing increased
remarkably to approximately 120 nm during the six haze episodes, compared with the
GMD of 50 nm under clean conditions. Two stages were identified: an initial rising
stage and a sustained increase stage. In the initial stage, the GMD of aerosols increased
by 50–60 nm for several hours, and the GMD then remained at 100–120 nm for several
days in the subsequent elevated pollution stage. This GMD increase during the initial
stage was mainly caused by the increase of accumulation-mode particles with diameters
of 100–1,000 nm and Aitken-mode particles (Fig. 10). Under clean conditions (SIA <
50 µg m$^{-3}$), the average contributions of the three modes (nucleation, Aitken, and
accumulation modes) to the number concentration were comparable, although the
number of nucleation-mode particles decreased with SIA concentration. Under light-
moderate pollution conditions (50 < SIA < 150 µg m$^{-3}$), the proportion of accumulation-
mode particles significantly increased from 35% to 60%, whereas the proportion of
Aitken-mode particles slowly decreased. As discussed in previous sections, regional
transport played a dominant role during the initial stage. This indicates that
condensation, coagulation, and chemical transformation in the transport pathway
increased the number of particles with a diameter range of 100–1,000 nm. Finally, the
contributions of Aitken-mode and accumulation-mode particles remained stable under
the heavy-pollution conditions (SIA >150 µg m$^{-3}$).
Aging processes play a critical role in the growth of particles during haze episodes.
According to observations, a significant coating of secondary components on BC was



found in the study period (Wang et al., 2018). Figure 11 presents a time series of the simulated $R_{BC}$, which is a favorable indicator of the aging degree (Oshima et al., 2009; Collier et al., 2018). Higher $R_{BC}$ indicates that BC had undergone a greater degree of aging. In this study, the simulated $R_{BC}$ was 2–10, with an average value of 4.5. This value is higher than that for fresh traffic source particles (Liu et al., 2017). Under pollution conditions, $R_{BC}$ was higher than that under clean conditions, with an average value of 5.0. $R_{BC}$ in Beijing even exceeded 10.0 in some extremely severe pollution events, which is close to observations of remote sites (Wang et al., 2017; Massoli et al., 2015) and aged particles (Cappa et al., 2012). Urban aerosols usually have a lower $R_{BC}$ because of fresh emissions and high $R_{BC}$ in this study indicates that Beijing aerosol particles were more aged during the haze episodes. On clean days, $R_{BC}$ ranged from 2 to 5, with an average of 2.8. This is similar to the $R_{BC}$ of vehicle emissions (<3) (Liu et al, 2017). Vehicle emissions contributed 70% of BC in downtown Beijing in 2016 after strict controls on coal burning had been implemented (Kebin He, personal communication).

Figure 12 shows the evolution of $R_{BC}$, the size distribution of number concentrations, and the GMD along the transport pathway from the source region to Beijing during the six haze episodes. The transport pathway was calculated using the HYSPLIT model. The figure clearly shows that the aerosol properties changed considerably along the transport pathway. In Ep1, the GMD of aerosols was only 97 nm at the initial site of the 24 h back trajectories ($T_{-24}$). At a larger transport distance, the diameters of aerosol particles were markedly increased to 128 nm in the middle ($T_{-12}$) and 134 nm at the final site ($T_0$) of the back trajectory. $R_{BC}$ increased from 3.6 at $T_{-24}$ to 8.7 at Beijing ($T_0$) because of BC being coated during the transport. This indicates that BC underwent considerable aging and increased in size while moving along the transport pathway; this would affect radiation and climate change (Cappa et al., 2012). Similar characteristics were discovered for Ep3–6. In Ep3, Ep4, Ep5, and Ep6, the GMD in Beijing ($T_0$) was 126, 117, 124, and 116 nm, respectively, compared with 96, 95, 99, and 111 nm in the middle point of transport ($T_{-12}$). $R_{BC}$ also increased to 4.6–7.6.



An exception was Ep2, in which the GMD (106 nm) and $R_{BC}$ (3.8) at the final ending
site (Beijing, $T_0$) were lower than those 6 h previously ($T_{-6}$). Regional transport
contributed 95% of BC at $T_{-6}$, whereas local emissions accounted for 87% of BC at $T_0$.
The number concentration was smaller at $T_{-6}$ than that at $T_0$. Therefore, we conclude
that regional transport of aged aerosols led to a high GMD at $T_{-6}$, and that the addition
of locally emitted fresh air caused a high number concentration but low GMD at $T_0$. In
clean areas, such as at $T_{-24}$ in Ep3 and Ep5, $R_{BC}$ was higher than 10 and the GMD was
considerably smaller.

## 4.4 Impact of heterogeneous chemistry on sulfate mass concentration

Current models generally account for a part of the observed $SO_4^{2-}$ concentrations
in China (Wang et al., 2014). Heterogeneous chemistry on aerosol surfaces under high
relative humidity has been considered a potential missing source of sulfate formation
(Cheng et al., 2016; Zheng et al., 2015; Li et al., 2017; Tang et al., 2016). Li et al. (2018)
developed a simple parameterization of heterogeneous chemistry and discovered that
$SO_2$ uptake on aerosols partly closed the gap between simulation and observation. In
their study, uptake coefficients were dependent on the aerosol core and shell species,
shell thickness, and amount of aerosol liquid water. Zheng et al. (2013) and Yang et al.
(2014) measured local source profiles, and they reported that primary sulfate from
industry and power plants accounted for a large fraction of PA.
In this study, we examined the contributions of gas ($SO_2$ + OH) and aqueous
chemistry, heterogeneous chemistry, and primary sulfate emissions to the sulfate mass
concentration in Beijing (Fig. 13). In Ep1, under high relative humidity, the contribution
of heterogeneous chemistry was 33%. Primary emissions exerted an effect mostly under
light to moderate pollution levels (sulfate <20 µg m$^{-3}$), whereas heterogeneous
chemistry played the largest role under high pollution levels (sulfate > 30 µg m$^{-3}$). The
contributions of gas and aqueous chemistry were largely consistent under all pollution



conditions (~30%). This indicates that high relative humidity and aerosol loading
accelerated the $SO_2$ chemical transformation. Interestingly, the contribution of
heterogeneous chemistry was markedly higher when the sulfate mass concentration
exceeded the threshold of 20 μg m$^{-3}$. Under high relative humidity and mass
concentration conditions, a higher aerosol surface area resulting from hygroscopic
growth provided a favorable media for heterogeneous reactions (Tie et al., 2017). The
aforementioned threshold is relatively similar to that during the haze episodes in the
winter of 2013 (Li et al, 2018). For policymakers, implementing measures to prevent
the sulfate concentration from exceeding this threshold is essential. Such measures
would be effective for avoiding extremely high sulfate levels. In other episodes,
heterogeneous chemistry was depressed because of the low relative humidity (<70%).
Gas and aqueous chemistry and primary emissions contributed 35%–40% and 58%–
61%, respectively. It should be noted that failure of the model to simulate mineral dust
led to underestimation of the sulfate level in Ep2. The interaction between $SO_2$ and
alkaline dust can contribute considerably to the sulfate concentration.

## 5 Conclusions


The contributions of regional transport to haze episodes over a megacity cluster
centered on Beijing have been under debate in recent decades. Investigating the
evolution of aerosol properties along the transport pathway may provide more
information on how researchers can improve the accuracy of regional transport and
chemistry impact assessments. To address one of the aims of the APHH 2016 winter
campaign, we employed a Eulerian chemical transport model (NAQPMS) and a
Lagrangian trajectory model (HYSPLIT) to assess the evolution of aerosols—in terms
of the number concentration, size distribution, and aging degree—in Beijing during six
haze episodes between November 15 and December 15, 2016. The transport of sulfate
and its precursors was also quantitatively investigated.
The results demonstrated that regional transport contributed 14%–31% to the
surface PM$_{2.5}$ mass concentration in Beijing during the six episodes, with a monthly



average contribution of 26%. Regarding aerosol components, 30%–62% of the SIA in
Beijing were regionally transported, whereas few PAs (<10%) were contributed from
emissions in other regions. Source regions differed between episodes. During high-
pollution periods, WHB, SD, and EHB were the main source regions of SIA regionally
transported to Beijing, whereas NHB and SX made greater contributions under clean
and light pollution conditions. This indicates the complexity of regional transport in
this megacity cluster.

The chemical transformation of $SO_2$ along the transport pathway from source

regions except Beijing to Beijing (RTC) was the major form of $SO_4^{2-}$ regional transport
and was more critical than the transport of sulfate formed in source regions except
Beijing (RTC). Compared with sulfate that was chemically transformed from Beijing-
emitted $SO_2$ and then blown back to Beijing (LTC), contribution of sulfate produced in
Beijing from Beijing-emitted $SO_2$ (LC) was generally greater. However, RTC markedly
increased in some episodes, and this explains the rapid formation of sulfate in Beijing.
This suggests that the joint control of local and regional $SO_2$ emissions is essential for
reducing the rapid formation of haze in this region.

Aerosols became considerably aged during transport in haze episodes, which

altered $R_{BC}$ and the size distribution of number concentrations. During haze episodes,
the GMD increased from less than 100 nm at the initial site to approximately 120 nm
at the final site (Beijing), and $R_{BC}$ increased from 2–4 to 4–8. The number of
accumulation-mode particles with a diameter range of 100–1,000 nm increased
considerably more than the number of particles of different modes. $R_{BC}$ in Beijing
during the episodes was higher than that of fresh particles (<1.5), which indicates that
BC in Beijing was more aged and thus more likely to affect radiation and climate.

Contributions from different pathways to sulfate in Beijing were also examined.

In episodes with high humidity (Ep1), the average contributions of gas and aqueous
chemistry, heterogeneous chemistry, and primary sulfate were comparable. Primary
emissions mostly had an effect under light to moderate pollution levels, whereas





heterogeneous chemistry played the most crucial role under high pollution levels. In other episodes (Ep2, Ep3, Ep4, Ep5, and Ep6), gas and aqueous chemistry and primary emissions contributed 35%–40% and 58%–61%, respectively.

**Acknowledgements:**

This work was supported by the Natural Science Foundation of China (41571130034; 91544227; 91744203; 41225019; 41705108) and the Chinese Ministry of Science and Technology (2018YFC0213205 and 2017YFC0212402).

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

**Figures**

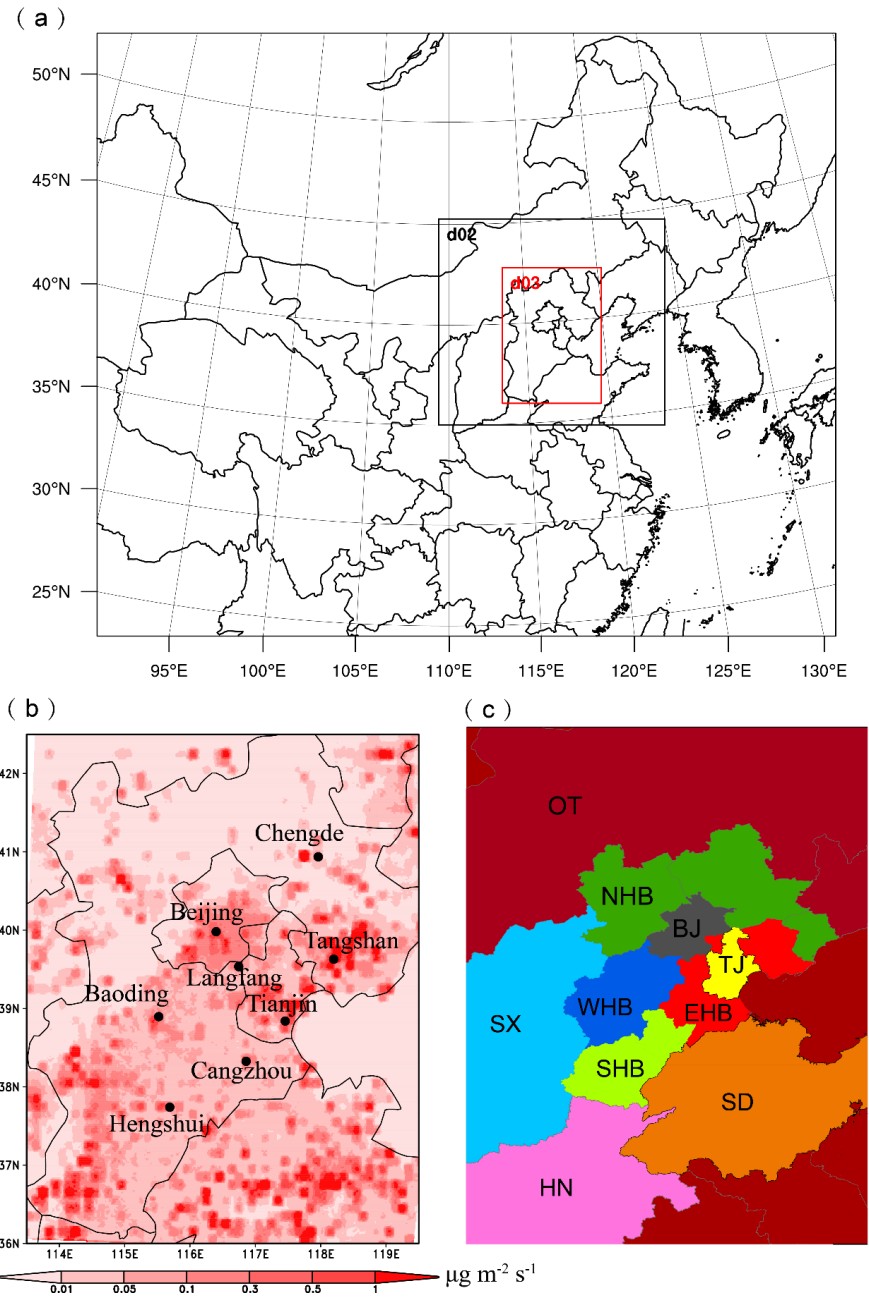


**Figure 1.** (a) Simulation domains. (b) Primary PM$_{2.5}$ emission rates of the innermost

domain and locations of observation sites (black dots). (c) tracer tagging regions which

are described in Table 1.






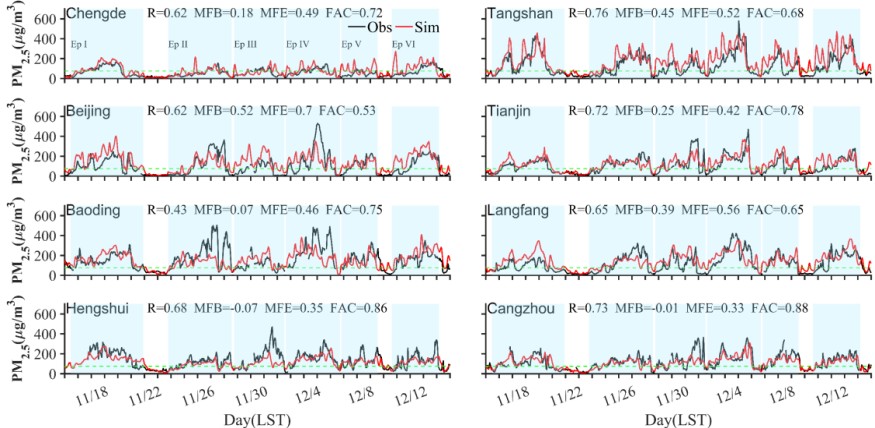

**Figure 2.** Comparison between the simulated and observed hourly concentrations of PM$_{2.5}$ for different sites. Black lines refer to observation and the red lines are simulation results; light blue shadows are six episodes identified.

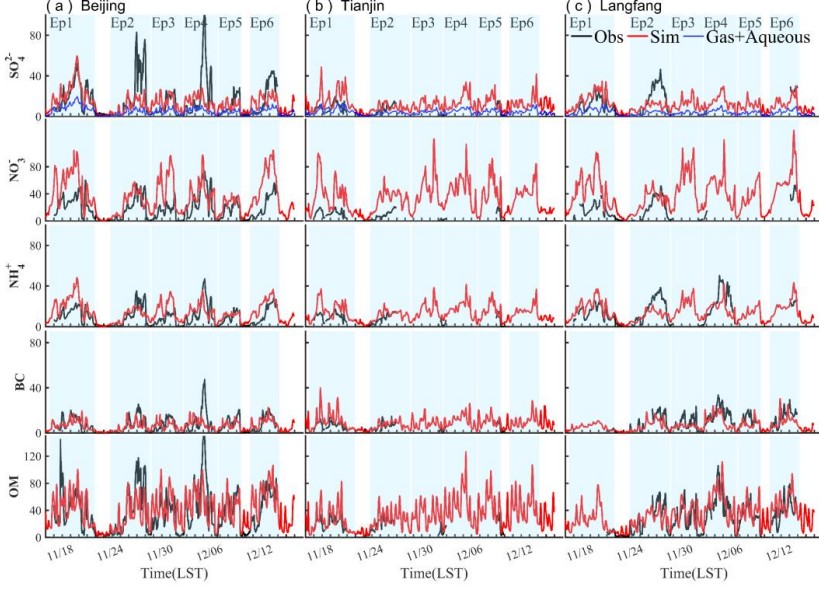

**Figure 3.** Comparison between the simulated (red) and observed (solid black) hourly components including sulfate, nitrate, ammonia, black carbon and organic aerosols at (a) Beijing, (b) Tianjin and (c) Langfang. Blue lines refer to sulfate produced by gas












and aqueous chemistry reactions.

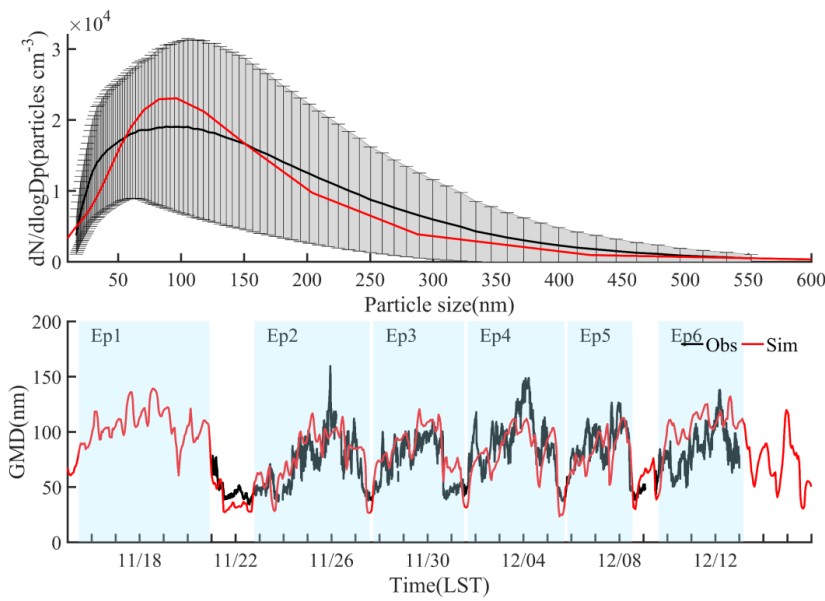


**Figure 4.** (a) Particle size distribution in Beijing at ground level. (b) Comparison of
geometric mean diameter (GMD) for particles during range of 16–600nm between
observation and simulation in Beijing. Black solid line and red solid line represent
observation and simulation respectively.

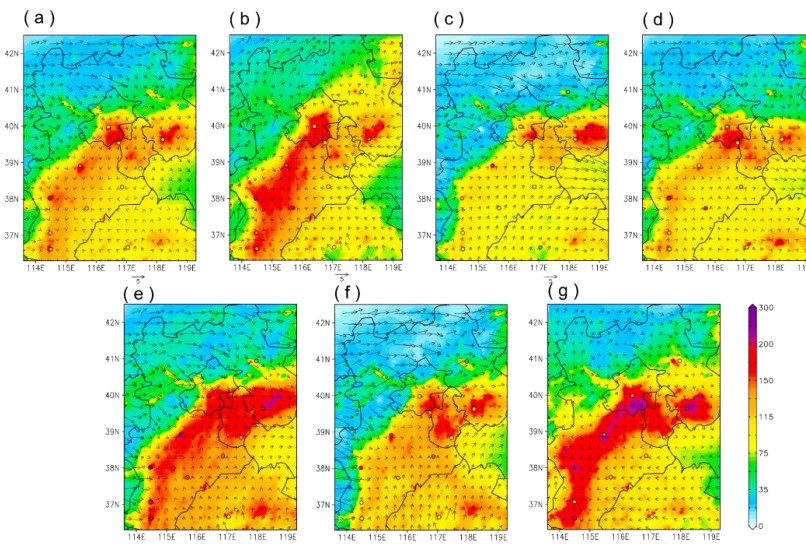


**Figure 5.** Spatial distribution of simulated average surface $PM_{2.5}$ ($\mu g\ m^{-3}$) and wind (m



s⁻¹) over BTH area. (a) average of whole study period, (b)–(g) episode average of
episode1−6 identified before. Solid circles represent observations with the same color
bar with simulations.

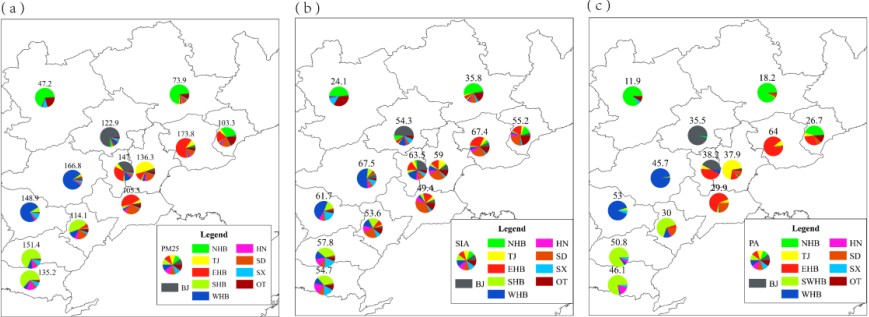

**Figure 6.** The contribution of regional transport and local emissions to the average (a)
total PM$_{2.5}$, (b) secondary inorganic aerosols (SIA), (c) primary aerosols (PA, BC and
primary PM$_{2.5}$) over BTH area. The numbers above the pie represent average
concentrations (μg m⁻³) of certain species in certain cities.

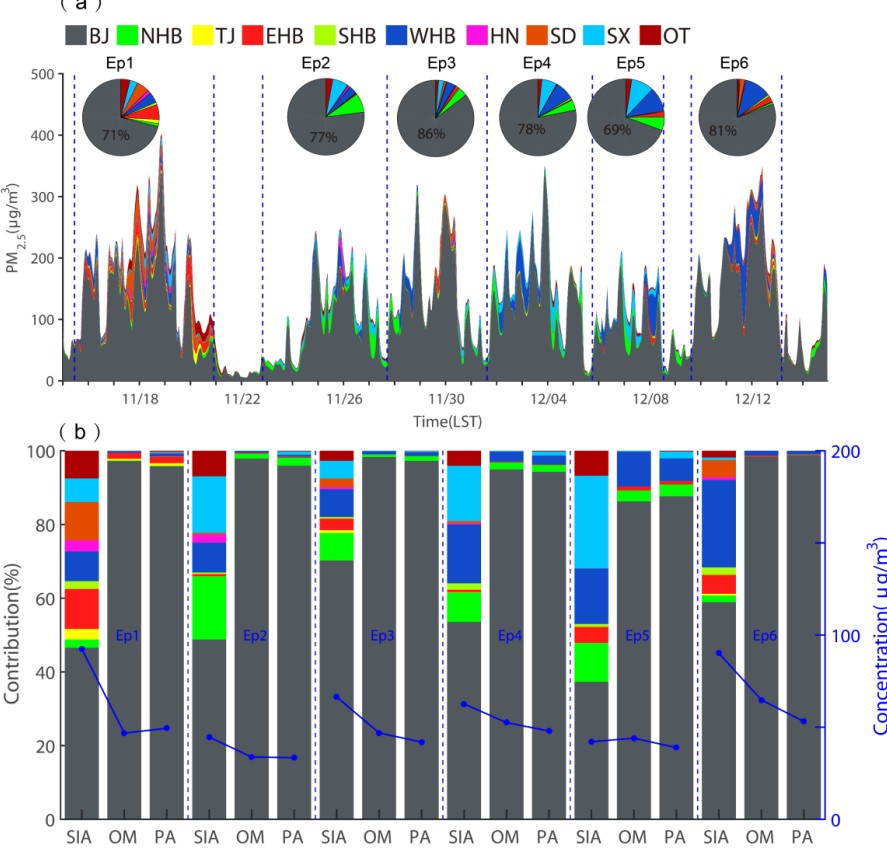

**Figure 7.** (a) Source contribution of PM$_{2.5}$ in Beijing and pies represent average status





of each episode; (b) Relative contribution of different regions to SIA, OM and PA in
Beijing at the surface layer during each episode (shaded). Concentrations are also
shown (blue line).


**Figure 8.** Relative contribution of regionally transported SIA under different levels of
pollution in Beijing during whole study period.

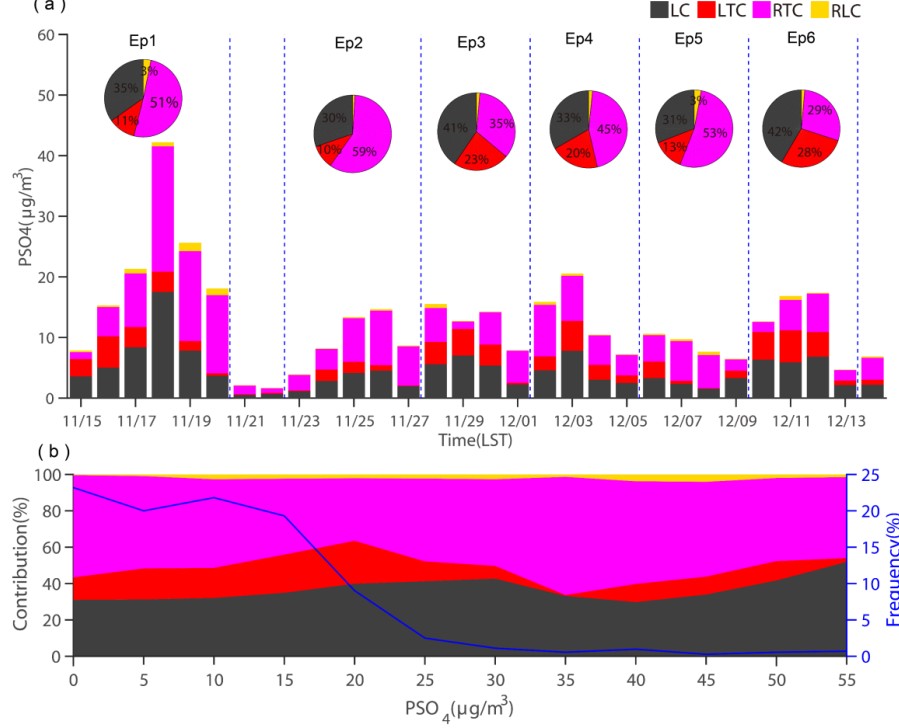


**Figure 9.** (a) Regional sources of chemical conversion of secondary sulfate in Beijing.





(b) Variation of region source of chemical conversion of secondary sulfate with hourly
surface sulfate concentration level in Beijing for the whole study period. LC means
sulfate locally produced from Beijing emitted $SO_2$; LTC refers to sulfate chemically
formed in regions except Beijing from the Beijing emitted $SO_2$; RTC is sulfate
chemically formed in the transport pathway to Beijing from $SO_2$ emitted in source
regions except Beijing; RLC is sulfate produced in regions except Beijing from locally
emitted $SO_2$ and transported to Beijing.

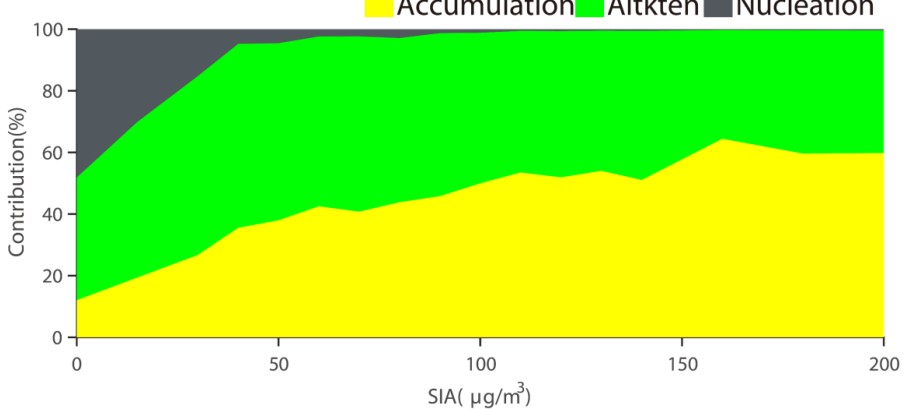

**Figure 10.** Variation of number concentration fraction of particles with SIA in Beijing
during whole study period.



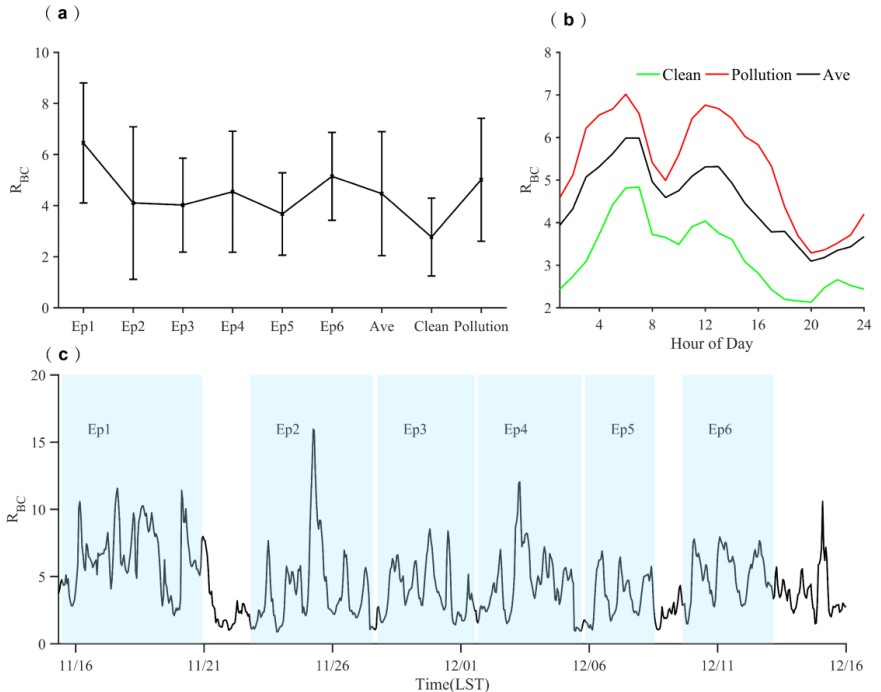

813

**Figure 11.** (a) average and standard variation of massing ratio of coating to BC ($R_{BC}$)

during different episodes and pollution levels, (b) diurnal cycles of $R_{BC}$ under different

pollution levels, (c) temporal variation of $R_{BC}$ during study period.

817

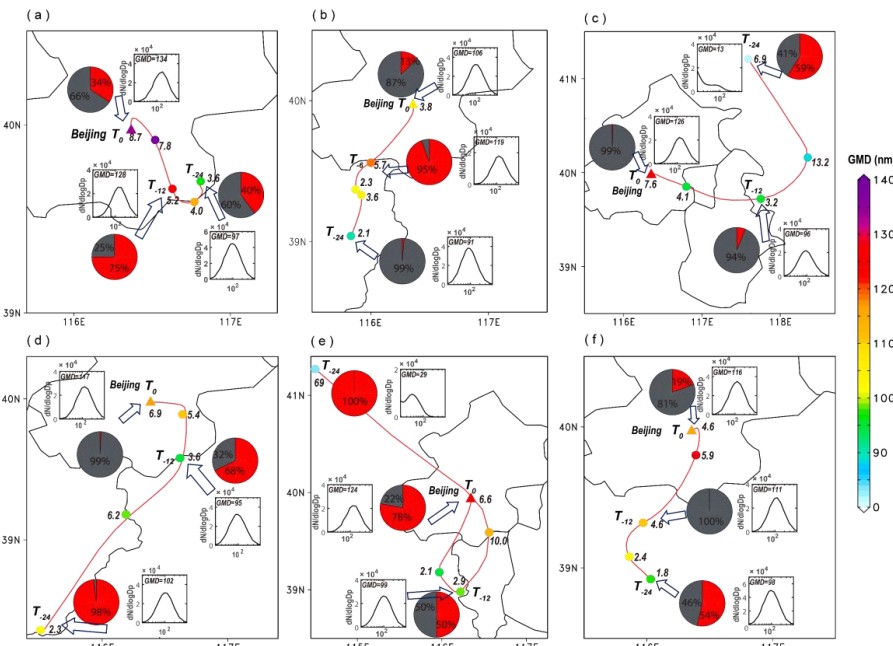

**Figure 12.** 24 h back trajectories for air mass at the altitude of 100 m and aerosol
properties along each trajectory. Panel a–f refers to episode 1–6 at 21:00 on November
18, 22:00 on November 25, 16:00 on November 29, 22:00 on December 03, 0:00 on
December 8, 22:00 on December 11 (LST). Triangles show ending site at Beijing, called
$T_0$. $T_{-6}$, $T_{-12}$, $T_{-18}$, $T_{-24}$ mean 6, 12, 18, 24 hours before arriving at ending site. The red
lines refer to backward trajectories and the solid shaded circles represent the geometric
mean particle size (GMD, nm) labeled in color bar on the right. The number beside the
solid circle is the mass ratio of coating to BC, called $R_{BC}$ for short. The pie chart shows
the region source of BC. The gray represents the local contribution, and the red
represents the contribution of regional transport. The blacklines refer to the distribution
of number concentration.



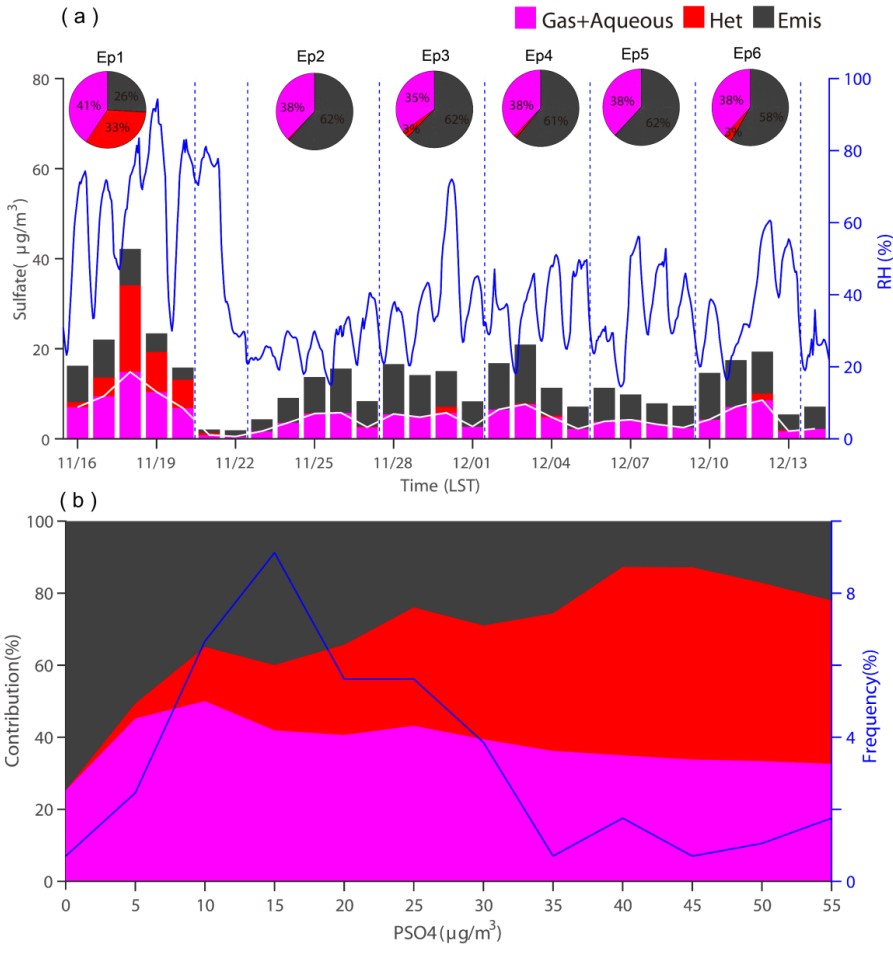

829

**Figure 13.** Contribution of different ways of sulfate formation in Beijing. (a) Daily
average. Blue line shows relative humidity at Beijing. Pies show average contribution
of different ways during each episode. (b) Relationship between sulphate concentration
and different pathways of sulphate formation during Ep1.


**Tables**

**Table 1.** Source-tagging regions and primary PM$_{2.5}$ emissions during 15 November–15
December, 2016 in this study. [a]

| Regions | Descriptions | Area | Population | GDP [b] | Emission [c] |
|---|---|---|---|---|---|
| | | $10^3$ km$^2$ | $10^6$ | $(10^{12}$ CNY) | $(10^9$g) |
| BJ | Beijing | 16.4 | 21.7 | 2.5 | 3.6 |
| TJ | Tianjin | 11.9 | 15.6 | 1.8 | 3.9 |
| NHB | Chengde, Zhangjiakou and | 84.1 | 11.6 | 0.4 | 3.6 |



| | | | | | | |
|---|---|---|---|---|---|---|
| | | Qinhuangdao | | | | |
| BTH | WHB | Baoding and Shijiazhuang | 38.0 | 21.2 | 0.9 | 8.1 |
| | EHB | Tangshan, Langfang and Cangzhou | 33.9 | 20.3 | 1.1 | 10.1 |
| | SHB | Hengshui, Xingtai and Handan | 33.3 | 22.9 | 0.7 | 6.8 |
| HN | | Henan | 167.0 | 95.3 | 4.0 | 26.6 |
| SD | | Shandong | 155.8 | 99.5 | 6.8 | 38.5 |
| SX | | Shanxi | 156.7 | 36.8 | 1.3 | 25.9 |
| OT | | Other regions | | | | |

[a] Regions are shown in Fig. 1c.
[b] GDP unit in 2016 is Chinese Yuan (CNY) (http://www.tjcn.org/tjgb/).
[c] $PM_{2.5}$ emissions data are obtained from the 2016 Multi–resolution Emission Inventory for China
(MEIC) with 0.25° × 0.25° resolution.