# Peer review of "Modeling of aerosol property evolution during winter haze"

_Atmospheric Chemistry and Physics, 2018_

## Referee Comment (RC1) · Anonymous Referee #1 · 17 Dec 2018

Modeling of aerosol property evolution during winter haze episodes over a megacity cluster in northern China: Roles of regional transport and heterogeneous reactions

Du et al.,

The study used NAPQPMS to re-produce the haze formation in North China Plain. They almost captured well the haze evolution and formation. Also, they did calculate the BC ageing processes in the haze formation in Beijing. Certainly, the model is very important to serve on haze formation in North China Plain. In particular, there are

lots of the measurement data from the APHH-BEIJING which can further improve the model accuracy. As the result from this study, they struggled to understand how the sulfate formation and BC aging, although they still not be prefect on reproducing it. As the novel of this paper, I might suggest one minor revision. (1) Seemly, the authors have some wrong citation in this paper. For example, L61 Tang et al,. (2016), In the reference, there are two Tang et al, (2016). I don't know which one should be cited here. L75-77, SIA mixed with BC. Not just revealed by Wang et al., (2018). There are different methods revealing it. You should pointed out it..Such as Wang et al., ESTL, (2017) 4 (11), 487-493; Peng et al., PNAS, 113(16), 4266-4271; (2) The study mostly considered the model could not re-produce the sulfate concentration although they did good nitrate and ammonium. Seemly, the authors think that the heterogeneous reactions should be dominant and missed very much in the model. I don't deny the claim. ALSO, the author should considered the primary sulfate emissions from the sources. As the recent study indicate the primary sulfate particles can be emitted from the household coal emissions. As I knew, the inventory from the household in rural areas still not good enough in the model. The authors should not miss the point in this study. For example, Zhang et al., JGR, 123 (22), 12,964-912,979. They found the coal burning in household can emit certain amounts of sulfates. (3) L24 Discovered, should change to revealed (4) L92-97. Introduction part, the description of participant should be avoid here. Probably, you can cite this paper. Shi et al, acp-2018-922, in ACPD. (5) L99, Sources of? Deleted of (6) L177 miss space before data You mentioned several times mixing states. I think that you should explain the mixing state in the model. Is this mixing state similar to Li et al, 2018, JGR, 121 (22), 13,784-713,798 or similar to Riemer et al., ACP, 13(22), 11,423–11,439. Seemly, they have different understanding on mixing state. How is your model you think? (7) L269-270. I don't think this is right reference here. The study worked on iron associated with ocean production. They didn't work on any aerosol particle in Beijing. (8) L321, along with (9) L379-389, I STILL want to emphasize the primary emissions of sulphates here., Seemly, you missed household emission in L500-502. (10) Section 4.4, the authors

tried to understand the heterogeneous reactions. The possible reactions should occur on aqueous layer which is related to the particle phase anymore (Sun et al., JGR, 123 (2), 1234-1243; Kuang et al., GRL, 43, 8744–8750.). Also, in 513-515. (11) Here I still confused on the aqueous chemistry and heterogeneous chemistry. What are differences in the model? Could you please list them. Seemly, heterogeneous chemistry happened in aqueous layer of particles. (12) Figure 6, the pie is too small to see clearly them (13) Figure 12. The figure is not clear for me. (14) Figure 13 what is PSO4? In X-axis

---

## Referee Comment (RC2) · Anonymous Referee #2 · 16 Jan 2019

The authors present results from a model study exploring the mass concentration levels of aerosol, in particular sulfate, over Northern China. They conclude that large fractions of secondary inorganic aerosol (SIA) are formed from SO2 that has been emitted from Beijing and that is oxidized in other regions and then transported back to the study region. Heterogeneous chemistry contributes a large fraction of this sulfate. In addition to SIA levels, they also look at the aging degree of aerosols, i.e. by the coating of black carbon particles (BC). In general, measurements and model results agree well. The topic of this manuscript is of high interest since sources of aerosol that leads to strong

haze periods in China are not fully understood yet. However, the study is in large parts obscure and details of the model are not explained which makes it difficult to appreciate the potentially new findings. The manuscript may be acceptable for publication after my comments below will be addressed.

Major comments

1) Only the levels and formation of sulfate are discussed whereas aerosol mass is composed of many more inorganic and organic compounds. While a comprehensive analysis of all aerosol constituents might exceed the scope of the paper, the limitation to sulfate should be made clear in the title, abstract and throughout the manuscript.

2) Several parts of the paper seem disconnected from each other and/or available information is not sufficiently used in the discussion:

a) The transport of SO2 away from Beijing and its subsequent oxidation followed by transport back to Beijing is an interesting thought. However, it would be much more convincing if HYSPLIT trajectories were included in the discussion showing this re-circulation of air masses.

b) Can the different source regions during the various episodes be connected to back trajectories and different emissions in the various source regions?

3) It is not clear what is exactly meant by 'aqueous' and 'heterogeneous' chemistry. Does aqueous phase chemistry only include cloud chemistry? Which oxidants are considered? Is metal-assisted oxidation included? Is heterogeneous chemistry the oxidation of S(IV) by NO2 or are other processes included as well? Which parameters are included in the model parameterization? How well are they constrained?

4) Most of the figures need to be improved. Contrasts are hard to see and often they are overloaded with information in much too small font.

Figure 2: What are the horizontal green lines?

Figure 4: Why is only the range up to 600 nm considered here even though the measurements and model bins extended further?

Figure 5: The solid circles are too small. Are they supposed to be colored as the caption suggests?

Figure 6: The pie charts and numbers are too small. Also the legends should be increased for better readability.

Figure 7: The black numbers on the dark grey pie charts are hard to read.

Figure 9: The text says that the figure shows regional sources. However, here all sources LC, LTC, RTC and RLC are shown.

Figure 11a): I suggest removing the lines between the symbols as they are meaningless and imply a non-existing trend.

Figure 12: - This figure contains way too much information. The numbers in the pie charts cannot be read. At the very least, this figure needs to be increased in size. However, it might be easier to include some of the information in an additional table.

- In the caption, it is not clear what 'black lines' are referred to here.

5) I am confused about the treatment of organics in the discussion of measured and modeled aerosol. It is well known that also organics can be directly emitted from various sources. However, for example, in the caption of Figure 7, OM and PA are separately listed. Please explain somewhere what PA (primary aerosol) includes and how the proportions of primary vs secondary organic aerosol are tracked within the model.

6) Many of the results seem trivial. They should be other presented as such or their novelty should be better highlighted if they indeed are surprising for the particular conditions in the current study.

a) l. 344 ff: It is well known that secondary aerosol exceeds primary aerosol after a short period of aging.

b) l. 340/341: This sentence is trivial. What other sources could SIA have if not chemical production?

c) It is mentioned that particles are aged within ~2 hours. Thus, is it surprising that in most cities a large fraction of PA originated from local emissions (l. 331)?

d) R(BC) for fresh particles should be ~0. Thus, the sentence in l. 457 is trivial.

e) l. 565: 'heterogeneous chemistry played the most crucial role under high pollution levels': It is not clear how you arrive at this conclusion. It is obvious that under high pollution levels (i.e. high SO2 levels) the contribution of PA might be small. However, it is not evident to me why the absolute contributions of gas and aqueous phase chemistry should not be enhanced equally.

Minor comments

l. 158 ff: It is not clear what ith here means. Do you mean 'emission from region i'?

l. 171: Is 'n ' the number of all regions. Please specify.

l. 279: Figure S2 only shows SO2, not NO2.

l. 364/365: I do not understand this sentence.

l. 368 – 374: This text sounds awkward and should be reworded. As it is written it implies that the clean or polluted conditions, respectively, determined the various source regions. However, it would be more reasonable to say that the wind direction from the various source regions led to the transport of the respective air masses into the study region. Because of the transport distance and/or pollution level in the source region, the resulting pollution level in the study region was high or low, respectively.

l. 478: '. . . this would affect radiation and climate change' should be removed.

l. 488: What does such a high R(BC) in a source region mean? Aerosol transported from that region will always appear aged.

l. 545: 'the major form of SO42-' should be replaced by 'the major source of SO42-'

Table S1: The caption should include more details.

Technical comments

l. 45: 'experiencing' should be 'experienced'

l. 71: 'physicochemical' misspelled

l. 204: Draxler misspelled

l. 235: were obtained

l. 296: 'respectively' misspelled
* * *

---

## Author Comment (AC1) · 17 Mar 2019

We thank the referees for the helpful comments. We have revised the manuscript according to the suggestions and responded to their concerns below. And we carefully edited the language and expression.

**Rerfee # 1**

**Comments from referee (1)** Seemly, the authors have some wrong citation in this paper. For example, L61 Tang et al., (2016), In the reference, there are two Tang et al, (2016). I don't know which one should be cited here. L75-77, SIA mixed with BC. Not just revealed by Wang et al., (2018). There are different methods revealing it. You should point out it. Such as Wang et al., ESTL, (2017) 4 (11), 487-493; Peng et al., PNAS, 113(16), 4266-4271;

**Response:** We will label a and b to distinguish the two references in the manuscript and corresponding modifications will be made in references.

As mentioned by the refree, there are different methods revealing mixing state of BC and SIA. Transmission electron microscopy (TEM) was used to calculate variation of fractal dimension (Df) to reflect soot aging process (Wang et al., 2017a). Also, change in the mass equivalent diameter ($\Delta D_{me}$) and coating fraction (ratio of variation in BC mass equivalent diameter to initial BC diameter, $\Delta D_{me}/D_{me,0}$) were also used to depict morphology variation and aging process (Peng et al., 2016). We will point out it and add these references to the manuscript.

**Changes in the manuscript:**

References have been corrected. Please refer to Page 3 line 60, Page 20 line 544.

Black carbon (BC) is usually more thickly coated by SIA and organic aerosols in transported and aged air masses than in fresh particles, as indicated by higher fractal dimension (Wang et al., 2017a), larger coating fraction (ratio of variation in BC mass equivalent diameter to initial BC diameter, $\Delta D_{me}/D_{me,0}$) (Peng et al., 2016) and higher mass ratio of coating to BC ($R_{BC}$) (Wang et al., 2018a). Please refer to Page 3 line 77-79.

**Comments from referee (2)** The study mostly considered the model could not reproduce the sulfate concentration although they did good nitrate and ammonium. Seemly, the authors think that the heterogeneous reactions should be dominant and missed very much in the model. I don't deny the claim. ALSO, the author should consider the primary sulfate emissions from the sources. As the recent study indicate the primary sulfate particles can be emitted from the household coal emissions. As I

knew, the inventory from the household in rural areas still not good enough in the model. The authors should not miss the point in this study. For example, Zhang et al., JGR, 123 (22), 12,964-12,979. They found the coal burning in household can emit certain amounts of sulfates.

**Response:** We agree with the point that primary sulfate particles emitted from the household coal emissions should be considered. Researchers found that high-maturity coals emitted sulfate (Zhang et al., 2018). Traditionally, model takes 5% of $SO_2$ emissions as primary sulfate emission according to convention. For another, residential sector lack reliable statistics and local emission factor measurements (Zhang et al., 2009), let alone the property of coals.

In the study, we take the primary sulfate emission into consideration. We took 40%, 6% and 15% of primary $PM_{2.5}$ from industrial, power and residential emissions, respectively, as primary sulfate emissions. And the percentages are taken according to observed source profiles from industrial, power and household coal-fired boiler (Wang et al., 2009; Zheng et al., 2013; Cao et al., 2014; Ma et al., 2015).

**Changes in the manuscript:**

We will add "household emission" and references in the manuscript. Please refer to Page 20 line 550-552 and Page 9 line 239-246.

**Comments from referee (3)** L24 Discovered, should change to revealed

**Response:** We will change "discovered" to "revealed" in the manuscript.

**Changes in the manuscript:** Changed.

**Comments from referee (4)** L92-97. Introduction part, the description of participant should be avoided here. Probably, you can cite this paper. Shi et al, acp-2018-922, in ACPD.

**Response:** The description of participants will be deleted. We will cite "(Shi et al., 2018)" to introduce the field campaign

**Changes in the manuscript:** "Details can be seen in Shi et al. (2018)" has been added here. Please refer to Page 4 line100.

**Comments from referee (5) L99, Sources of? Deleted of**

**Response:** Accepted. We will delete "of" in the manuscript.

**Changes in the manuscript:** Deleted. Please refer to Page 4 Line 105.

**Comments from referee (6)** L177 miss space before data. You mentioned several times mixing states. I think that you should explain the mixing state in the model. Is this mixing state similar to Li et al, 2016, JGR, 121 (22), 13,784-13,798 or similar to Riemer et al., ACP, 13(22), 11,423–11,439. Seemly, they have different understanding on mixing state. How is your model you think?

**Response:** Agree. Thanks for your suggestion.

There are different understandings about mixing state. Transmission electron microscopy (TEM) was used to investigate the mixing structures of individual particles (Li et al., 2016). And a particle that contains two or more components can be regarded as internally mixed. Otherwise, it is considered to be externally mixed. An affine ratio of the average per-particle species diversity and the bulk population species diversity were used to quantitatively present aerosol population mixing state, namely degree to which population is externally mixed versus internally mixed (Riemer and West, 2013).

In this study, mixing state is a description of aerosol population, taking microphysics process into consideration. Mixing state is assumed to be semi-external (Yu and Luo, 2009;Chen et al., 2014), includes internal mixing, external mixing and core-shell mixing. Seeding particles, including BC, OC, dust, sea salt are generated by emission and secondary aerosols formed by nucleation. Then seeding particles are coated by secondary particles including sulfate, nitrate, ammonia and secondary organic aerosols through condensation, coagulation, chemistry reactions, equilibrium uptake and hygroscopic growth process. Nucleated secondary particle is internally mixed while primary particles coated with SIA or SOA are considered as core-shell mixing. And these coated particles are external mixing with each other. Mixing states in the model are as shown below.

[Figure]

Figure 1 Microphysical process and mixing state in the model

**Changes in the manuscript:** We will add description of mixing state in the manuscript. Please refer to Page 6 line 156-158.

**Comments from referee** (7) L269-270. I don't think this is right reference here. The study worked on iron associated with ocean production. They didn't work on any aerosol particle in Beijing.

**Response:** Thanks for your suggestion. We use this paper to show that heterogenous chemistry can happen on mineral dust surface. Wang et al. (2017b; 2018b) confirmed the importance of dust heterogeneous reactions, and we will add these references here.

**Changes in the manuscript:** References have been changed. "As discussed in the last paragraph, Beijing had high mineral loadings for Ep2, which provided a favorable medium for chemical transformation of anthropogenic $SO_2$ into sulfate in the form of $CaSO_4$ or $MgSO_4$ (Wang et al., 2018b;Wang et al., 2017b). " Please refer to Page 11 line 297.

**Comments from referee** (8) L321, along with

**Response:** Thanks for your suggestion. We think about it carefully. We mean pollution accumulates at the foot of the mountain and forms pollution belts along the mountains because of the obstruction of the terrain. "along" is a preposition here, so we think maybe it's better to use "along"

**Changes in the manuscript:** The sentence will stay the same here.

**Comments from referee** (9) L379-389, I STILL want to emphasize the primary emissions of sulphates here., Seemly, you missed household emission in L500-502.

**Response:** Thanks for your suggestion. L379-389 is meant to tell the different understanding and definition of regional transport. And the transport of secondary inorganic aerosols includes transport of both its precursors and secondary aerosols. Household burning was found emits a large amount of sulfate (Zhang et al., 2012;Zhang et al., 2018). And we will add "household emission" after L502.

**Changes in the manuscript:** Household emission will be added. Please refer to Page 20 line 551-552.

**Comments from referee** (10) Section 4.4, the authors tried to understand the heterogeneous reactions. The possible reactions should occur on aqueous layer which is related to the particle phase anymore (Sun et al., JGR, 123 (2), 1234-1243; Kuang et al., GRL, 43, 8744–8750.). Also, in 513-515.

**Response:** Thanks for your suggestion. We agree.

Researchers investigated deliquescent phenomena of aerosols and found ammonium sulfate played significant roles (Sun et al., 2018;Kuang et al., 2016). And their results focused on phase transition of particles, especially haze particles under high RH(60–80%) with aqueous shell. This phase transition process is not also considered in detail by model.

In the model, growth factor and aerosol water are calculated by thermodynamic equilibrium model ISORROPIA. As mechanism of heterogenous chemistry is not fully understood, heterogeneous chemistry is commonly parameterized using a pseudo-first-order rate constant (Jacob, 2000). And reaction rate is as follows:

$$\kappa = \left[ \frac{r}{D} + \frac{4}{C\gamma} \right]^{-1} \times A$$

$\gamma$ is the ratio of the number of collisions that result in reaction to the theoretical total number of collisions, and it depends in general on temperature and types of aerosols. A is the surface area. Under high relative humidity, a higher aerosol surface area resulting from hygroscopic growth is favorable for heterogeneous reactions.

**Changes in the manuscript:**

Please refer to Page 20 line 565-566.

**Comments from referee** (11) Here I still confused on the aqueous chemistry and heterogeneous chemistry. What are differences in the model? Could you please list them. Seemly, heterogeneous chemistry happened in aqueous layer of particles.

**Response:** Thanks for your suggestion. Descriptions about heterogeneous chemistry in the paper may not be so clear. We will explain the treatment of aqueous and heterogeneous reactions in the model.

In the model, aqueous chemistry happened only in **cloud water**, which is provided by WRF. But cloud does not occur frequently in winter over Beijing-Tianjin-Hebei area. Aqueous chemistry reactions mainly include the following five reactions:

$$H_2O_2 + SO_2 \rightarrow SO_4^{2-}$$

$$O_3 + SO_2 \rightarrow SO_4^{2-}$$

$$MHP^a + SO_2 \rightarrow SO_4^{2-}$$

$$PAA^b + SO_2 \rightarrow SO_4^{2-}$$

$$O_2 + SO_2 + Fe(III)/Mn(II) \rightarrow SO_4^{2-}$$

[a] MHP: methylhydroperoxide; [b] PAA: peroxyacetic acid.

Heterogeneous chemistry reactions happen on **aerosol surface** and are related with aerosol liquid water which is provided by ISORROPIA. Heterogeneous chemistry is commonly parameterized using a pseudo-first-order rate constant and reaction rate is as follows:

$$k_{ij} = [\frac{r_{ij}}{D_j} + \frac{4}{C_j\gamma_j}]^{-1} \times A_{ij}$$

$k_{ij}$ is the reaction rate of $j^{th}$ reaction in the $i^{th}$ bin. $\gamma$ is the uptake coefficient for reactant j, $A_{ij}$ is the surface area of the $i^{th}$ bin of reactant j, $r_{ij}$ is the radium of the $i^{th}$ bin of reactant j, $D_j$ is the gas phase molecular diffusion coefficient for reactant j. $C_j$ is molecular speed of reactant j in gas phase.

In this study, we also add the heterogeneous chemistry of SO₂ to the model, and more details can be seen in (Li et al., 2018).

$$SO_2 + Aerosol \rightarrow SO_4^{2-}$$

Uptake coefficient is calculated as the following equation:

$$\gamma_{SO2} = \begin{cases} 1 \times 10^{-4} & awc \geq C_{up} \\ (awc - C_{low}) \times \dfrac{10^{-4} - 10^{-6}}{C_{up} - C_{low}} & C_{low} < awc < C_{up} \\ 1 \times 10^{-6} & awc \leq C_{low} \end{cases}$$

$\gamma_{SO2}$ is the uptake coefficient of SO₂. Assuming that the upper limit of $\gamma_{SO2}$ does not exceed uptake coefficient on dust surface, upper limit of $\gamma_{SO2}$ is $10^{-4}$ and lower limit is $10^{-6}$. $C_{low}$, the lower limit of AWC required by heterogeneous chemistry, is 10 μg m⁻³. When uptake coefficient reaches peak, $C_{up}$, the upper limit of AWC is 300 μg m⁻³.

**Changes in the manuscript:** Differences between aqueous chemistry and heterogeneous chemistry in the model will be added to the manuscipt. Please refer to Page 5 line 131-133.

**Comments from referee** (12) Figure 6, the pie is too small to see clearly them

**Response:** Thanks for your suggestion. We will modify the pies in figure 6.

**Changes in the manuscript:** Please refer to Page 33 line 868.

[Figure]

Figure 6. The contribution of regional transport and local emissions to the average (a) total PM$_{2.5}$, (b) secondary inorganic aerosols (SIA), (c) primary aerosols (PA, BC and primary PM$_{2.5}$) over BTH area. The numbers above the pie represent average concentrations ($\mu g\ m^{-3}$) of certain species in certain cities.

**Comments from referee** (13) Figure 12. The figure is not clear for me.

**Response:** Thanks for your suggestion. Figure 12. shows variation of aerosol properties (region sources, aging degree, geometric mean diameter and number concentration) along transport. We will improve the figure and include some of the information in an additional table.

**Changes in the manuscript:** Please refer to Page 40 line 903-914.

[Figure]

Figure 12. Variation of aerosol properties along transport. Panel a–f refers to episode 1–6. The red lines refer to 24 h backward trajectories. Aerosol properties include geometric mean diameter (GMD [nm], red numbers), mass ratio of coating to BC ($R_{BC}$, the black numbers beside the solid blocks, an indicator of aging degree), region source of BC (pies, the red represents regional transport and the gray is the local contribution). Shaded triangles are ending points of back trajectories and shaded circles are points along trajectories per six hours. T-6, T-12, T-18, T-24 mean 6, 12, 18, 24 hours before arriving at ending site. Ending times of backtrajectories are before pollution peaks at 21:00 on November 18, 22:00 on November 25, 16:00 on November 29, 22:00 on December 03, 0:00 on December 8 and 22:00 on December 11 (LST), respectively.

Table 1. Aerosol properties along transport, including geometric mean diameter (GMD [nm]), mass ratio of coating to BC ($R_{BC}$), number concentration (N) and contribution of region source to BC (Cr [%]). $T_0$ means ending points of back trajectories and $T_n$ means n hours before arriving at the ending point.

| | | $T_{-24}$ | $T_{-18}$ | $T_{-12}$ | $T_{-6}$ | $T_0$ |
|---|---|---|---|---|---|---|
| **Ep1** | $R_{BC}$ | 3.6 | 4.0 | 5.2 | 7.8 | 8.7 |
| | GMD | 97 | 115 | 128 | 139 | 134 |
| | N | 28994 | 15494 | 15204 | 15592 | 19242 |
| | Cr | 40 | 93 | 75 | 7 | 34 |
| **Ep2** | $R_{BC}$ | 2.1 | 3.6 | 2.3 | 5.7 | 3.8 |
| | GMD | 91 | 104 | 102 | 119 | 106 |
| | N | 23909 | 15189 | 17961 | 10994 | 20121 |
| | Cr | 1.2 | 0.14 | 0.01 | 95 | 13 |
| **Ep3** | $R_{BC}$ | 6.9 | 13.2 | 3.2 | 4.1 | 7.6 |
| | GMD | 13 | 74 | 96 | 95 | 126 |
| | N | 22234 | 11880 | 13481 | 14241 | 12945 |
| | Cr | 59 | 81.4 | 6.2 | 8.8 | 1 |
| **Ep4** | $R_{BC}$ | 2.3 | 6.2 | 3.6 | 5.4 | 6.9 |
| | GMD | 102 | 98 | 95 | 111 | 117 |
| | N | 19754 | 12805 | 21116 | 10536 | 17199 |
| | Cr | 98 | 56 | 68 | 25 | 1 |
| **Ep5** | $R_{BC}$ | 69 | 10.0 | 2.9 | 2.1 | 6.6 |
| | GMD | 29 | 114 | 99 | 95 | 124 |
| | N | 8617 | 8086 | 16494 | 28211 | 13696 |
| | Cr | 100 | 100 | 50 | 4 | 78 |
| **Ep6** | $R_{BC}$ | 1.8 | 2.4 | 4.6 | 5.9 | 4.6 |
| | GMD | 98 | 103 | 111 | 129 | 116 |
| | N | 31691 | 23691 | 17885 | 12897 | 21955 |
| | Cr | 54 | 0.17 | 0.01 | 65 | 19 |

**Comments from referee (14)** Figure 13 what is PSO4? In X-axis

**Response:** Thanks for your suggestion. PSO4 refers to sulfate concentration here. We will change it into sulfate to keep consistent with Figure 13a.

**Changes in the manuscript:** Please refer to Page 43 line 928.

[Figure]

Figure 13. Contribution of different ways of sulfate 830 formation in Beijing. (a) Daily average. Blue line shows relative humidity at Beijing. Pies show average contribution of different ways during each episode. (b) Relationship between sulphate concentration and different pathways of sulphate formation during Ep1.

**Major comments 2:** Several parts of the paper seem disconnected from each other and/or available information is not sufficiently used in the discussion:

a) The transport of $SO_2$ away from Beijing and its subsequent oxidation followed by transport back to Beijing is an interesting thought. However, it would be much more convincing if HYSPLIT trajectories were included in the discussion showing this recirculation of air masses.

**Response:** Thanks for your suggestion. HYSPLIT trajectories are good indicators of recirculation of air masses.Figure blow shows that air masses flow away from Beijing and finally blown back to Beijing at different time (02:00 on November 17, 11:00 on November 29, and 23:00 on December 12 [LST]). Note that the trajectories were conducted at a single location in Beijing. And at those moments, LTC (sulfate chemically formed in regions except Beijing from the Beijing emitted $SO_2$) accounts for large part of sulfate (figure 9a). Results are quiet consistent.

[Figure]

Figure 1 36 h backward trajectories at different start time (02:00 on November 17, 11:00 on November 29, and 23:00 on December 12 [LST]) at Beijing.

**Changes in the manuscript:** We will add trajectories to the manuscript as supplement. Please refer to Page 16 line 442-445.

b) Can the different source regions during the various episodes be connected to back trajectories and different emissions in the various source regions?

Response: Yes. Back trajectories during various episodes are shown below.

During Ep1, airflow passed through Hebei province to Beijing, then through Tangshan, the Bohai sea, and finally flown back to Beijing. Western Hebei and eastern Hebei are the mainly source regions except local emission. Figure 7b showed that EHB and WHB accounted for 11% and 8% of SIA at Beijing, respectively. During Ep2, airmass flown from Inner Mongolia through Shanxi, NHB, WHB and arrived at Beijing. During Ep3, airflow travelled from Inner Mongolia through NHB, EHB, SD and finally arrived at Beijing. Figure 7b showed that SX, NHB, EHB and SD accounted for 5%, 7%, 3% and 2% of SIA at Beijing, respectively. During Ep5, airflow travelled from Inner Mongolia through SX, NHB, WHB and stayed in Baoding for some time, then finally arrived at Beijing. These are quite consistent with the result of SIA shown in Figure 7b. During Ep6, airmass mainly came from Shandong, through SHB, WHB and finally arrived at Beijing. What's more, the height of trajectory within WHB is low, so contribution of WHB should be big, which agreed with results of Figure 7b, WHB contributed 24% to SIA at Beijing during Ep6.

As back trajectory model did not consider chemical conversion of precursors, so the results are not exactly the same with those of 3-D air quality model. But in general, the on-line source-tagged module results agreed well with those from the backward trajectories.

[Figure]

Figure 2 72 h backward trajectories during different episodes (05:00 on November 20, 18:00 on November 26, 23:00 on November 29, 20:00 on December 03, 05:00 on December 08 and 11:00 on December 12 [LST]) at Beijing at 100m. Line color represents height of trajectories. a-f refer to Ep1-6.

**Changes in the manuscript:** Back trajectories during various episodes will added to the manuscript as supplement. Please refer to Page 14 line 396-401.

**Major comments 3:** It is not clear what is exactly meant by 'aqueous' and 'heterogeneous' chemistry. Does aqueous phase chemistry only include cloud chemistry? Which oxidants are considered? Is metal-assisted oxidation included? Is heterogeneous chemistry the oxidation of S(IV) by NO₂ or are other processes included as well? Which parameters are included in the model parameterization? How well are they constrained?

**Response:** In this study, aqueous phase chemistry only includes cloud chemistry. Oxidants of aqueous chemistry include $O_3$, $H_2O_2$, methyl hydroperoxide and peroxyacetic acid. Metal-assisted oxidation is also included.

Heterogeneous chemistry includes oxidation of S(IV) on aqueous layer of aerosols. Heterogeneous chemistry is parameterized according to the scheme of Li et al. (2018), using a pseudo-first-order rate constant and reaction rate is as follows

$$\kappa = \left[\frac{r}{D} + \frac{4}{C\gamma}\right]^{-1} \times A$$

$\gamma$ is the ratio of the number of collisions that result in reaction to the theoretical total number of collisions, and it depends in general on temperature and species.

A is the surface area.

$\gamma$ is the most important parameter, and it is related to aerosol liquid water (awc), and amounts of oxidants (Li et al., 2018) . The uptake coefficient of $SO_2$ is as follows.

$$\gamma_{SO2} = \begin{cases} 1 \times 10^{-4} & awc \geq C_{up} \\ (awc - C_{low}) \times \dfrac{10^{-4} - 10^{-6}}{C_{up} - C_{low}} & C_{low} < awc < C_{up} \\ 1 \times 10^{-6} & awc \leq C_{low} \end{cases}$$

$\gamma_{SO2}$ is the uptake coefficient of $SO_2$. Assuming that the upper limit of $\gamma_{SO2}$ does not exceed uptake coefficient on dust surface, upper limit of $\gamma_{SO2}$ is $10^{-4}$ and lower limit is $10^{-6}$. $C_{low}$, the lower limit of AWC required by heterogeneous chemistry, is 10 µg m$^{-3}$. When uptake coefficient reaches peak, $C_{up}$, the upper limit of AWC is 300 µg m$^{-3}$. More details can be found in Li et al., (2018).

**Changes in the manuscript:** Please refer to Page 5 line 131-133 and Page 6 line 166-168.

**Major comments 4:** Most of the figures need to be improved. Contrasts are hard to see and often they are overloaded with information in much too small font.

Figure 2: What are the horizontal green lines?

**Response:** Green lines mean 75 µg m$^{-3}$, as a criterion judging whether pollution or not.

**Changes in the manuscript:** We will add this in the caption and manuscript. Please refer to Page 30 line 849-850.

"**Figure 2.** Comparison between the simulated and observed hourly concentrations of PM$_{2.5}$ for different sites. Black lines refer to observation and the red lines are simulation results; light blue shadows are six episodes identified; green lines mean 75 μg m$^{-3}$, as a criterion judging whether pollution or not."

Figure 4: Why is only the range up to 600 nm considered here even though the measurements and model bins extended further?

**Response:** As the upper limitation of observation is about 600nm. What's more, number concentration of PM$_{2.5}$ is almost within 600nm (Du et al., 2017;Liu et al., 2016). So only the range up to 600 nm was considered.

Figure 5: The solid circles are too small. Are they supposed to be colored as the caption suggests?

**Response:** Yes, the solid circles are colored with the same bar with the simulation. We will increase the circles for better readability.

**Changes in the manuscript:** Please refer to Page 31 line 861.

[Figure]

**Figure 5.** Spatial distribution of simulated average surface PM$_{2.5}$ (µg m$^{-3}$) and wind (m s$^{-1}$) over BTH area. (a) average of whole study period, (b)–(g) episode average of episode1–6 identified before. Solid circles represent observations with the same color bar with simulations.

Figure 6: The pie charts and numbers are too small. Also the legends should be increased for better readability.

**Response: Accepted. We will change it in the manuscript.**

**Changes in the manuscript:** Please refer to Page 33 line 868.

[Figure]

**Figure 6.** The contribution of regional transport and local emissions to the average (a) total PM$_{2.5}$, (b) secondary inorganic aerosols (SIA), (c) primary aerosols (PA, BC and primary PM$_{2.5}$) over BTH area. The numbers above the pie represent average concentrations ($\mu$g m$^{-3}$) of certain species in certain cities.

Figure 7: The black numbers on the dark grey pie charts are hard to read.

**Response:** Accepted. We will change the color in the manuscript.

**Changes in the manuscript:** Please refer to Page 35 line 874.

[Figure]

**Figure 7.** (a) Source contribution of PM₂.₅ in Beijing and pies represent average status of each episode; (b) Relative contribution of different regions to SIA, OM and PA in Beijing at the surface layer during each episode (shaded). Concentrations are also shown (blue line).

Figure 9: The text says that the figure shows regional sources. However, here all sources LC, LTC, RTC and RLC are shown.

**Response:** We will change the text in the manuscript.

**Changes in the manuscript:** "(a) Sources of secondary sulfate in Beijing." Please refer to Page 37 line 886-887.

Figure 11a): I suggest removing the lines between the symbols as they are meaningless and imply a non-existing trend.

**Response:** We will revise the figure in the manuscript

**Changes in the manuscript:** Please refer to Page 38 line 898.

[Figure]

Figure 11. (a) average and standard variation of massing ratio of coating to BC ($R_{BC}$) during different episodes and pollution levels, (b) diurnal cycles of $R_{BC}$ under different pollution levels, (c) temporal variation of $R_{BC}$ during study period.

Figure 12: - This figure contains way too much information. The numbers in the pie charts cannot be read. At the very least, this figure needs to be increased in size.

However, **it might be easier to include some of the information in an additional table.**

- In the caption, it is not clear what 'black lines' are referred to here.

**Response:** Accepted. We will improve the figure and more details can be seen in Table1.

**Changes in the manuscript:** Please refer to Page 40 line 903-914

[Figure]

Figure 12. Variation of aerosol properties along transport. Panel a–f refers to episode 1–6. The red lines refer to 24 h backward trajectories at the altitude of 100 m. Aerosol properties include geometric mean diameter (GMD [nm], red numbers beside the solid blocks), mass ratio of coating to BC ($R_{BC}$, the black numbers beside the solid blocks, an indicator of aging degree), region source of BC (pies, the red represents regional transport and the gray is the local contribution). Shaded triangles are ending points of back trajectories, called $T_0$. Shaded circles are points along trajectories per six hours. $T_{-6}$, $T_{-12}$, $T_{-18}$, $T_{-24}$ mean 6, 12, 18, 24 hours before arriving at ending site. Ending times of backtrajectories are before pollution peaks at 21:00 on November 18, 22:00 on November 25, 16:00 on November 29, 22:00 on December 03, 0:00 on December 8 and 22:00 on December 11 (LST), respectively.

Table 1. Aerosol properties along transport, including geometric mean diameter (GMD [nm]), mass ratio of coating to BC ($R_{BC}$), number concentration (N) and contribution of region source to BC (Cr [%]). $T_0$ means ending points of back trajectories and $T_n$ means n hours before arriving at the ending point.

| Period | Property | $T_{-24}$ | $T_{-18}$ | $T_{-12}$ | $T_{-6}$ | $T_0$ |
|--------|----------|-----------|-----------|-----------|----------|-------|
| Ep1 | $R_{BC}$ | 3.6 | 4.0 | 5.2 | 7.8 | 8.7 |
| | GMD | 97 | 115 | 128 | 139 | 134 |
| | N | 28994 | 15494 | 15204 | 15592 | 19242 |
| | Cr | 40 | 93 | 75 | 7 | 34 |
| Ep2 | $R_{BC}$ | 2.1 | 3.6 | 2.3 | 5.7 | 3.8 |
| | GMD | 91 | 104 | 102 | 119 | 106 |
| | N | 23909 | 15189 | 17961 | 10994 | 20121 |
| | Cr | 1.2 | 0.14 | 0.01 | 95 | 13 |
| Ep3 | $R_{BC}$ | 6.9 | 13.2 | 3.2 | 4.1 | 7.6 |
| | GMD | 13 | 74 | 96 | 95 | 126 |
| | N | 22234 | 11880 | 13481 | 14241 | 12945 |
| | Cr | 59 | 81.4 | 6.2 | 8.8 | 1 |
| Ep4 | $R_{BC}$ | 2.3 | 6.2 | 3.6 | 5.4 | 6.9 |
| | GMD | 102 | 98 | 95 | 111 | 117 |
| | N | 19754 | 12805 | 21116 | 10536 | 17199 |
| | Cr | 98 | 56 | 68 | 25 | 1 |
| Ep5 | $R_{BC}$ | 69 | 10.0 | 2.9 | 2.1 | 6.6 |
| | GMD | 29 | 114 | 99 | 95 | 124 |
| | N | 8617 | 8086 | 16494 | 28211 | 13696 |
| | Cr | 100 | 100 | 50 | 4 | 78 |
| Ep6 | $R_{BC}$ | 1.8 | 2.4 | 4.6 | 5.9 | 4.6 |
| | GMD | 98 | 103 | 111 | 129 | 116 |
| | N | 31691 | 23691 | 17885 | 12897 | 21955 |
| | Cr | 54 | 0.17 | 0.01 | 65 | 19 |

**Major comments 5**: I am confused about the treatment of organics in the discussion of measured and modeled aerosol. It is well known that also organics can be directly emitted from various sources. However, for example, in the caption of Figure 7, OM and PA are separately listed. Please explain somewhere what PA (primary aerosol) includes and how the proportions of primary vs secondary organic aerosol are tracked within the model.

**Response:**

In the Observation, Organic matters were measured by using an Aerodyne high-resolution time-of-flight aerosol mass spectrometer at Beijing (Sun et al., 2015). The OC/EC in aerosol was measured by a field semi-online OC/EC analyzer from Sunset Laboratory Inc. (USA) with a PM2.5 cyclone inlet at Tianjin and Lang Fang (Gao et al., 2016).

In the model, organics include two parts: primary organic aerosols (POA) and secondary organic aerosols (SOA). POA are from emission of OC. Six secondary organic aerosols (SOA), in which two were from anthropogenic precursors (toluene and higher aromatics) and four were from biogenic precursors (monoterpene and isoprene), were explicitly treated in NAQPMS. The formation of SOA consists of two steps: the photooxidation of volatile organic compounds (VOCs) by OH to produce semi-volatile organic compounds (SVOCs); then SVOCs can be partioned between gaseous/liquid or gaseous/solid to form secondary organic aerosols.

In the study, PA means sum of non-organic primary $PM_{2.5}$ and BC. OM includes primary organic aerosols and secondary organic aerosols.

POA is tracked by OC. SOA is tracked by its precursors.

**Changes in the manuscript:**

The measure of organics was clearly stated. Please refer to Page 9-10 line 253-254, 258-260.

PA means sum of non-organic primary $PM_{2.5}$ and BC. Please refer to Page 13 line 355.

OM includes primary organic aerosols and secondary organic aerosols. Please refer to Page 14 line 392.

**Major comments 6**: Many of the results seem trivial. They should be other presented as such or their novelty should be better highlighted if they indeed are surprising for the particular conditions in the current study.

a) l. 344 ff: It is well known that secondary aerosol exceeds primary aerosol after a short period of aging.

**Response:** Yes, after a period of aging, secondary aerosols exceed primary aerosols. Precursors of secondary aerosols can stay one day to one week in the air. We quantified the sources of SIA and found that SIA mainly came from regional transport. Therefore, region-joint control of precursors is necessary. And this is consistent with observation result that after region-joint control measures taken, derease ratio of SIA was bigger than that of PA.

**Changes in the manuscript:** Regional controls "on precursors" was added. Please refer to Page 14 line 373.

b) l. 340/341: This sentence is trivial. What other sources could SIA have if not chemical production?

Response: Accepted.

**Changes in the manuscript:** we will delete this sentence. Please refer to Page 14 line 370.

c) It is mentioned that particles are aged within _2 hours. Thus, is it surprising that in most cities a large fraction of PA originated from local emissions (l. 331)?

**Response:** Aging of BC within 2 hours means hydrophobic BC particles can be converted to be hydrophilic within 2 hours. BC, $NO_2$ and $SO_2$ are often from ths same source in anthropogenic source region. Aged means aerosols coated by secondary inorganic and organic aerosols through condensation and coagulation in the air. Therefore, BC may be aged shortly after emission in the anthropogenic source area .

Region sources of PA are managed by tagging emission regions. BC aging could not directly change sources although their life time can be changed due to different wet deposition efficiency.

d) R(BC) for fresh particles should be _0. Thus, the sentence in l. 457 is trivial.

**Response:** Yes. This sentence will be deleted. Comparisons with other countries and other regions in China show that $R_{BC}$ of Beijing is higher than that in Fresno, located in vally region of California, of 2-3.5 (Collier et al., 2018). $R_{BC}$ of Beijing in this study is smaller than that in Tibet Plateau with average of about 7.7 (Wang et al., 2017). But $R_{BC}$ under pollution is close to SOA dominated BC-coating particles with $R_{BC}$ of 8 (Lee, 2017).

**Changes in the manuscript:** we will delete this sentence. Please refer to Page 18 line 505.

e) l. 565: 'heterogeneous chemistry played the most crucial role under high pollution levels': It is not clear how you arrive at this conclusion. It is obvious that under high pollution levels (i.e. high $SO_2$ levels) the contribution of PA might be small. However, it is not evident to me why the absolute contributions of gas and aqueous phase chemistry should not be enhanced equally.

**Response:**

UV radiative transfer (TUV) model, which calculates the effect of aerosol, cloud, and gaseous pollutant on photolysis, was included in NAQPMS (Li et al., 2011). Under pollution levels, radiation decreases and OH concentration is lower, and production of gas chemistry reduced.

In the model, aqueous phase chemistry only included in cloud chemistry.

Reaction rate of heterogeneous is positively related to **aerosol surface area and uptake coefficient**. Aerosol size increases due to hygroscopic growth under high relative humidity, which provides a good medium for heterogeneous reactions. Also, uptake coefficient is related to humidity, and higher relative humidity during high pollution levels is favorable for gas uptake. Therefore, absolute contributions of heterogeneous increased but contribution of gas and aqueous chemistry decreased.

Therefore, heterogeneous chemistry will play a relatively crucial role under high pollution levels.

**Changes in the manuscript:** 'heterogeneous chemistry played a relatively crucial role under high pollution levels during Ep1'. Please refer to Page 22 line 619-620.

**Minor comments**

l. 158 ff: It is not clear what ith here means. Do you mean 'emission from region i'?

**Response:** Yes. We will add "region" to this line.

**Changes in the manuscript:** Please refer to Page 7 line 176,178.

l. 171: Is 'n' the number of all regions. Please specify.

**Response:** Yes, 'n' is the number of all regions. We will specify it in the manuscript.

**Changes in the manuscript:** Please refer to Page 7 line 191.

l. 279: Figure S2 only shows SO2, not NO2.

**Response:** We will change it.

**Changes in the manuscript:** Please refer to Page 11 line 308.

l. 364/365: I do not understand this sentence.

**Response:** This sentence is not closely related to stated before and it will be deleted.

**Changes in the manuscript:** This sentence will be deleted. Please refer to Page 14 line 395.

l. 368 – 374: This text sounds awkward and should be reworded. As it is written it implies that the clean or polluted conditions, respectively, determined the various source regions. However, it would be more reasonable to say that the wind direction from the various source regions led to the transport of the respective air masses into the study region. Because of the transport distance and/or pollution level in the source region, the resulting pollution level in the study region was high or low, respectively.

**Response:** We will change it in the manuscript. Variation of wind direction with SIA level will be added to the manuscript.

When Beijing is controlled by strong northerly wind, NHB and SX are the main source regions, contributing up to 30% and 19%, resulting in clean conditions (SIA < 50 μg m$^{-3}$). When Beijing is mainly affected by southerly wind (southeast, south and southwest), WHB, EHB and SD become the main source regions, contributing 27%, 13% and 15%, respectively. Strong emissions of source regions lead to heavier pollution level in Beijing. When Beijing is dominated by weak southeast wind, contribution from far regions like HN and SD increases. Continuous transport and accumulation lead to severe pollution (SIA > 150 μg m$^{-3}$).

**Changes in the manuscript:** Please refer to Page 15 line 402-417.

[Figure]

**Figure 8.** (a) Relative contribution of regionally transported SIA under different pollution levels in Beijing during whole study period; (b)Variation of wind direction under different pollution levels in Beijing during whole study period.

l. 478: ': : : this would affect radiation and climate change' should be removed.

**Response:** Accepted.

**Changes in the manuscript:** Please refer to Page 19 line 527.

l. 488: What does such a high R(BC) in a source region mean? Aerosol transported from that region will always appear aged.

**Response:** At $T_{-24}$ in Ep5, simulated $R_{BC}$ is up to 69, which is much higher than that in anthropogenic source regions. At that time, air mass is in Zhangjiakou, a clean city and almost all BC is from regional transport. GMD is 29 nm and number concentration is cm$^{-3}$, relatively small compared with other time of transport. Therefore, high $R_{BC}$ is caused by high concentration of transported coated aerosols. Correspondingly, in polluted areas, the aging degree may be relatively low due to continuous BC emissions.

Our model can quantify mass ration of coating to BC, that's $R_{BC}$. As the refree points out that aerosol aged during transport, but aging degree varies in different locations and during different episodes. And aging degree can affect extinction properties of aerosols. This index is studied seldomly by 3-D air quality models before. And the importance of studying about aging degree is confirmed by laboratory and researches (Liu et al., 2017).

l. 545: 'the major form of SO42-' should be replaced by 'the major source of SO42-'

**Response:** We will change it in the manuscript.

**Changes in the manuscript:** Please refer to Page 22 line 597.

Table S1: The caption should include more details.

**Response:** We will change it in the manuscript.

**Changes in the manuscript:**

Table S1. Statistics performances of meteorological simulations, including temperature and relative humidity at 2 m, and wind speed at 10 m. Statistical parameters include correlation coefficient (R), Normalized Mean Bias (NMB) and Root Mean Squared Error (RMSE).

**Technical comments**

l. 45: 'experiencing' should be 'experienced'

**Response:** We will change it in the manuscript.

**Changes in the manuscript:** Please refer to Page 2 line 45.

l. 71: 'physicochemical' misspelled

**Response:** We will change it in the manuscript.

**Changes in the manuscript:** Please refer to Page 3 line 72.

l. 204: Draxler misspelled

**Response:** We will change it in the manuscript.

**Changes in the manuscript:** Please refer to Page 8 line 224.

l. 235: were obtained

**Response:** We will change it in the manuscript.

**Changes in the manuscript:** Please refer to Page 10 line 264.

l. 296: 'respectively' misspelled

**Response:** We will change it in the manuscript.

**Changes in the manuscript:** Please refer to Page 12 line 325.

[revised manuscript text omitted]

Figure S6 36 h backward trajectories at different start time (02:00 on November 17,

11:00 on November 29, and 23:00 on December 12 [LST]) at Beijing.

**Table S1.** Statistics performances of meteorological simulations, including
temperature and relative humidity at 2 m, and wind speed at 10 m. Statistical parameters
include correlation coefficient (R), Normalized Mean Bias (NMB) and Root Mean
Squared Error (RMSE).

|  |  | Obs | Sim | NMB | R | RMSE |
|---|---|---|---|---|---|---|
| **T2(K)** | Beijing | 2.42 | 3.31 | 0.37 | 0.88 | 2.11 |
|  | Tianjin | 4.05 | 3.73 | -0.08 | 0.93 | 1.48 |
|  | Langfang | 2.13 | 3.06 | 0.44 | 0.89 | 2.13 |
|  | Chengde | -3.39 | -1.66 | -0.51 | 0.89 | 3.31 |
| **RH2(%)** | Beijing | 53.93 | 39.53 | -0.27 | 0.69 | 21.87 |
|  | Tianjin | 56.95 | 47.58 | -0.16 | 0.73 | 18.09 |

| | | | | | | |
|---|---|---|---|---|---|---|
| | Langfang | 60.39 | 46.54 | -0.23 | 0.71 | 21.54 |
| | Chengde | 61.56 | 55.96 | -0.09 | 0.47 | 20.26 |
| | Beijing | 1.68 | 1.93 | 0.15 | 0.65 | 1.30 |
| | Tianjin | 1.76 | 2.46 | 0.39 | 0.70 | 1.50 |
| **WS10(m/s)** | Langfang | 1.23 | 2.15 | 0.74 | 0.57 | 1.56 |
| | Chengde | 1.16 | 1.41 | 0.21 | 0.63 | 1.22 |

**Table S2.** Aerosol properties along transport, including geometric mean diameter (GMD [nm]), mass ratio of coating to BC ($R_{BC}$), number concentration (N) and contribution of region source to BC (Cr [%]). $T_0$ means ending points of back trajectories and $T_n$ means n hours before arriving at the ending point.

| Period | Property | $T_{-24}$ | $T_{-18}$ | $T_{-12}$ | $T_{-6}$ | $T_0$ |
|---|---|---|---|---|---|---|
| | $R_{BC}$ | 3.6 | 4.0 | 5.2 | 7.8 | 8.7 |
| **Ep1** | GMD | 97 | 115 | 128 | 139 | 134 |
| | N | 28994 | 15494 | 15204 | 15592 | 19242 |
| | Cr | 40 | 93 | 75 | 7 | 34 |
| | $R_{BC}$ | 2.1 | 3.6 | 2.3 | 5.7 | 3.8 |
| **Ep2** | GMD | 91 | 104 | 102 | 119 | 106 |
| | N | 23909 | 15189 | 17961 | 10994 | 20121 |
| | Cr | 1.2 | 0.14 | 0.01 | 95 | 13 |
| | $R_{BC}$ | 6.9 | 13.2 | 3.2 | 4.1 | 7.6 |
| **Ep3** | GMD | 13 | 74 | 96 | 95 | 126 |
| | N | 22234 | 11880 | 13481 | 14241 | 12945 |
| | Cr | 59 | 81.4 | 6.2 | 8.8 | 1 |
| | $R_{BC}$ | 2.3 | 6.2 | 3.6 | 5.4 | 6.9 |
| **Ep4** | GMD | 102 | 98 | 95 | 111 | 117 |
| | N | 19754 | 12805 | 21116 | 10536 | 17199 |
| | Cr | 98 | 56 | 68 | 25 | 1 |
| | $R_{BC}$ | 69 | 10.0 | 2.9 | 2.1 | 6.6 |
| **Ep5** | GMD | 29 | 114 | 99 | 95 | 124 |
| | N | 8617 | 8086 | 16494 | 28211 | 13696 |
| | Cr | 100 | 100 | 50 | 4 | 78 |
| | $R_{BC}$ | 1.8 | 2.4 | 4.6 | 5.9 | 4.6 |
| **Ep6** | GMD | 98 | 103 | 111 | 129 | 116 |
| | N | 31691 | 23691 | 17885 | 12897 | 21955 |

| Cr | 54 | 0.17 | 0.01 | 65 | 19 |

---

## Author Response (AR2)

We thank the referees for the helpful comments. We have revised the manuscript according to the suggestions and responded to their concerns below.

**Comments from referee (1)** l. 35: Remove 'these changes would affect regional radiation and climate' as these consequences are not discussed in the manuscript.

**Response:** Accepted. We will delete it.

**Changes in the manuscript:** The sentence will be deleted. Please refer to Page 2 Line 34-35.

**Comments from referee (2)** l. 95-97: 'The heterogeneous reaction parameters are rarely related to the key parameters such as mixing state and aerosol water contents in previous studies.' - This sentence is not clear. Do you mean that there is not much data available that describe the parameters for heterogeneous reactions as a function of aerosol composition and water content? Please add a reference that shows these dependencies and lack of data.

I suggest rewording 'For many heterogeneous reactions, comprehensive data sets for the reaction parameters as a function of aerosol composition and water content are not available.'

**Response:** Accepted. Studies found that uptake coefficients (γ) were dynamically related to aerosol liquid water and aerosol mix states (coating thickness) (Riemer et al., 2009; Morgan et al., 2015). Although these parameters are obtained in some observation studies (Bian et al., 2014; Zhang et al., 2016). The parameterization of heterogeneous has rarely been linked to these parameters (Morgan et al., 2015; Zheng et al., 2015; Li et al., 2018).

**Changes in the manuscript:**

Please refer to Page 4 Line 91-97.

**Comments from referee (3)** l. 132: Please specify the aqueous processes that are included. Is it only the oxidation reactions of S(IV) by $O_3$, $H_2O_2$, methyl hydroperoxide and peroxyacetic acid? Or does it also include additional reactions, e.g. the redox chemistry of metal ions - as your response to my previous review implies?

**Response:** Accepted. Aqueous chemistry reactions include the oxidation reactions of S(IV) by $O_3$, $H_2O_2$, methyl hydroperoxide, peroxyacetic acid and redox chemistry of metal ions, $Fe^{3+}$ and $Mn^{2+}$.

**Changes in the manuscript:**

In the model, aqueous chemistry happens only in cloud water, including the oxidation reactions of S(IV) by $O_3$, $H_2O_2$, methyl hydroperoxide, peroxyacetic acid and oxidation catalysis of transition metal ions ($Fe^{3+}$ and $Mn^{2+}$).

Please refer to Page 5 Line 129-132.

**Comments from referee (4)** l. 134: 'are related with aerosol water'. I suggest adding some more details of the calculation of γSO2 as it is not intuitive that an uptake coefficient depends on the aerosol water content. For example:

'Assuming that the upper limit of $\gamma SO2$ does not exceed the uptake coefficient on dust surfaces, the upper limit of $\gamma SO2$ is 10-4 for AWC > 300 µg m-3 and its lower limit is 10-6 if AWC < 10 µg m-3. More details can be found in Li et al., (2018).'

**Response:** Accepted.

**Changes in the manuscript:**

Heterogeneous chemistry reactions happen on aerosol's surface with aqueous layer and are related with aerosol liquid water. For sulfate, assuming that the upper limit of $\gamma_{SO_2}$ does not exceed the uptake coefficient on dust surfaces, the upper limit of $\gamma_{SO_2}$ is $10^{-4}$ for aerosol water content (AWC) > 300 µg m$^{-3}$ and its lower limit is $10^{-6}$ if AWC < 10 µg m$^{-3}$. More details can be found in Li et al. (2018).

Please refer to Page 5 Line 133-136

**Comments from referee (5)** l. 217: 'heterogeneous chemistry to sulfate'- do you mean 'heterogeneous formation of sulfate'?

**Response:** Yes. We will change it.

**Changes in the manuscript:**

To quantitatively assess the contribution of primary emissions, traditional chemistry reactions (gas-phase and aqueous chemistry reactions), and heterogeneous formation of sulfate, three sensitivity simulations were conducted.

Please refer to Page 8 Line 219.

**Comments from referee (6)** l. 329: How is 'aging degree' quantified? There might be several measures that can be used to describe how aged a particle is, such as the coating thickness of BC or the oxidation degree of OA. However, to my knowledge, there is no universal model parameter that describes the 'aging degree'.

**Response:** Yes, there is no universal model parameter that describe the aging degree. In this study, the mass ratio of coating to BC ($R_{BC}$) was used as an indicator of aging degree. And $R_{BC}$ is a surrogate for coating thickness (Wang et al., 2018).

**Changes in the manuscript:**

Herein, the mass ratio of coating to BC ($R_{BC}$) was used as an indicator of aging degree, which has been widely used in previous studies (Oshima et al., 2009; Collier et al., 2018). Please refer to Page 12 Line 321-322.

**Comments from referee (7)** l. 368: 'mechanisms of PA…formation'. 'Formation' implies that these aerosols are chemically formed; however, PA (primary aerosols) are directly emitted and do not undergo any formation processes.

**Response:** Accepted.

**Changes in the manuscript:** The difference in source apportionment between PA and SIA was related to the emission of PA and formation mechanisms of SIA. Please refer to Page 13 Line 366-367.

**Comments from referee (8)** l. 393: I had already asked in my previous review about the classification of PA and OM. It should be clarified that PA only includes primary inorganic aerosol (perhaps name it PIA) as you clarified now that OM includes both primary and secondary organic aerosol.

**Response:** Accepted. We change all PA to PIA in the manuscript. And PIA was explained in detail, when first appeared in Page 13 Line 354.

**Changes in the manuscript:** We change all PA to PIA in the manuscript, such as, Page 13 line 359, 364, 366-369 and Figure 6 and 7.

**Comments from referee (9)** l. 407 and l. 409: Specify: …contributing up to 30% and 19% to total $PM_{2.5}$ levels?

**Response:** Accepted. It is contribution to SIA levels, and we will specify it.

**Changes in the manuscript:** When Beijing is controlled by strong northerly wind, NHB and SX are the main source regions, contributing up to 30% and 19% to SIA, resulting in clean conditions (SIA < 50 µg m$^{-3}$). When Beijing is mainly affected by southerly wind (southeast, south and southwest), WHB, EHB and SD become the main source regions, contributing up to 27%, 13% and 15% to SIA, respectively.
Please refer to Page 15 Line 403 and 405.

**Comments from referee (10)** l. 613: This sentence is hard to understand now. Do you mean the following?
- '$R_{BC}$ in Beijing during the episodes was higher than that of OTHER urban regions (Collier et al., 2018)'?
- At high pollution levels, $R_{BC}$ was close to that in remote regions (Wang et al., 2017a)

**Response:** Accepted.

**Changes in the manuscript:** $R_{BC}$ in Beijing during the episodes was higher than that of other urban regions (Collier et al., 2018). At high pollution levels, $R_{BC}$ was close to that in remote regions (Wang et al., 2017a). Please refer to Page 22 Line 603 and 604.

**Comments from referee (11)** l. 617-621: This text seems contradictory: 'In episodes with high humidity (Ep1), the average contributions of gas and aqueous chemistry, heterogeneous chemistry, and primary sulfate were comparable.'
'…heterogeneous chemistry played a relatively crucial role under high pollution levels during Ep1.'

**Response:** Accepted. The average contributions' magnitude of gas and aqueous chemistry, heterogeneous chemistry, and primary sulfate was comparable during episodes with high humidity (Ep1). But their relative contributions varied with pollution levels. Primary emissions mostly had an effect under light to moderate pollution levels, whereas heterogeneous chemistry played a relatively crucial role under high pollution levels during Ep1.

**Changes in the manuscript:** In episodes with high humidity (Ep1), the average contributions of gas and aqueous chemistry, heterogeneous chemistry, and primary sulfate were comparable. But their relative contributions varied with pollution levels. Under light to moderate pollution levels, primary emissions mostly had an effect. But under high pollution levels during Ep1, heterogeneous chemistry played a relatively crucial role. Please refer to Page 22 Line 606 and 610.

Also seen in Page 2 line 37.

**Technical comments**

l. 28: '…was the major form of $SO_4^{2-}$ regional transport' is unclear. Do you mean 'was the major source of $SO_4^{2-}$'?

**Response:** Accepted. Yes.

**Changes in the manuscript:** The chemical transformation of $SO_2$ in the transport pathway from source regions to Beijing was the major source of $SO_4^{2-}$. Please refer to Page 1 Line 27-28.

l. 44/45: 'has been experiencing' or 'has experienced'

**Response:** Accepted.

**Changes in the manuscript:** In past decades, a megacity cluster in China that is centered on Beijing and includes 28 cities (272,500 $km^2$, a population of 191.7 million people) has experienced frequent severe and persistent haze episodes (Zhao et al., 2013; Sun et al., 2014; Sun et al., 2016). Please refer to Page 2 Line 44.

l. 114: Detailed analysis of…

**Response:** Accepted.

**Changes in the manuscript:** Detailed analysis of the transport of precursors or secondary products, and heterogeneous reactions was mainly focused on sulfate, as recent studies indicated that sulfate is a key driver for severe haze events (Huang et al., 2014; Zheng et al., 2015). Please refer to Page 4 Line 111.

l. 356: 'non-organic' should be 'inorganic'

**Response:** Accepted.

**Changes in the manuscript:** Figure 6 shows the contributions of regional transport and local emissions to average $PM_{2.5}$, primary inorganic aerosol (PIA, BC and inorganic primary $PM_{2.5}$), and SIA levels in different cities during the study period. Please refer to Page 13 Line 354.

l. 443: 'convinced' seems to be a wrong word here; 'supported' might fit better.

**Response:** Accepted.

**Changes in the manuscript:** And recirculation of air masses can be supported by HYSPLIT trajectories (Fig. S6). Please refer to Page 16 Line 433.

l. 505: '…had undergone a greater degree of aging' sounds awkward. It should be, for example, 'had undergone more ageing' or 'had reached a higher degree of ageing'

**Response:** Accepted.

**Changes in the manuscript:** Higher $R_{BC}$ indicates that BC had undergone more ageing. Please refer to Page 18 Line 494-495.

[revised manuscript text omitted]

11:00 on November 29, and 23:00 on December 12 [LST]) at Beijing.

**Table S1.** Statistics performances of meteorological simulations, including temperature and relative humidity at 2 m, and wind speed at 10 m. Statistical parameters include correlation coefficient (R), Normalized Mean Bias (NMB) and Root Mean Squared

Error (RMSE).

|  |  | Obs | Sim | NMB | R | RMSE |
|---|---|---|---|---|---|---|
| **T2(K)** | Beijing | 2.42 | 3.31 | 0.37 | 0.88 | 2.11 |
|  | Tianjin | 4.05 | 3.73 | -0.08 | 0.93 | 1.48 |
|  | Langfang | 2.13 | 3.06 | 0.44 | 0.89 | 2.13 |
|  | Chengde | -3.39 | -1.66 | -0.51 | 0.89 | 3.31 |
| **RH2(%)** | Beijing | 53.93 | 39.53 | -0.27 | 0.69 | 21.87 |
|  | Tianjin | 56.95 | 47.58 | -0.16 | 0.73 | 18.09 |
|  | Langfang | 60.39 | 46.54 | -0.23 | 0.71 | 21.54 |

| | | | | | |
|---|---|---|---|---|---|
| | Chengde | 61.56 | 55.96 | -0.09 | 0.47 | 20.26 |
| **WS10(m/s)** | Beijing | 1.68 | 1.93 | 0.15 | 0.65 | 1.30 |
| | Tianjin | 1.76 | 2.46 | 0.39 | 0.70 | 1.50 |
| | Langfang | 1.23 | 2.15 | 0.74 | 0.57 | 1.56 |
| | Chengde | 1.16 | 1.41 | 0.21 | 0.63 | 1.22 |

**Table S2.** Aerosol properties along transport, including geometric mean diameter
(GMD [nm]), mass ratio of coating to BC ($R_{BC}$), number concentration (N) and
contribution of region source to BC (Cr [%]). $T_0$ means ending points of back
trajectories and $T_n$ means n hours before arriving at the ending point.

| Period | Property | $T_{-24}$ | $T_{-18}$ | $T_{-12}$ | $T_{-6}$ | $T_0$ |
|---|---|---|---|---|---|---|
| **Ep1** | $R_{BC}$ | 3.6 | 4.0 | 5.2 | 7.8 | 8.7 |
| | GMD | 97 | 115 | 128 | 139 | 134 |
| | N | 28994 | 15494 | 15204 | 15592 | 19242 |
| | Cr | 40 | 93 | 75 | 7 | 34 |
| **Ep2** | $R_{BC}$ | 2.1 | 3.6 | 2.3 | 5.7 | 3.8 |
| | GMD | 91 | 104 | 102 | 119 | 106 |
| | N | 23909 | 15189 | 17961 | 10994 | 20121 |
| | Cr | 1.2 | 0.14 | 0.01 | 95 | 13 |
| **Ep3** | $R_{BC}$ | 6.9 | 13.2 | 3.2 | 4.1 | 7.6 |
| | GMD | 13 | 74 | 96 | 95 | 126 |
| | N | 22234 | 11880 | 13481 | 14241 | 12945 |
| | Cr | 59 | 81.4 | 6.2 | 8.8 | 1 |
| **Ep4** | $R_{BC}$ | 2.3 | 6.2 | 3.6 | 5.4 | 6.9 |
| | GMD | 102 | 98 | 95 | 111 | 117 |
| | N | 19754 | 12805 | 21116 | 10536 | 17199 |
| | Cr | 98 | 56 | 68 | 25 | 1 |
| **Ep5** | $R_{BC}$ | 69 | 10.0 | 2.9 | 2.1 | 6.6 |
| | GMD | 29 | 114 | 99 | 95 | 124 |
| | N | 8617 | 8086 | 16494 | 28211 | 13696 |
| | Cr | 100 | 100 | 50 | 4 | 78 |
| **Ep6** | $R_{BC}$ | 1.8 | 2.4 | 4.6 | 5.9 | 4.6 |
| | GMD | 98 | 103 | 111 | 129 | 116 |
| | N | 31691 | 23691 | 17885 | 12897 | 21955 |
| | Cr | 54 | 0.17 | 0.01 | 65 | 19 |